There are amendments to this paper

# Cellular deconvolution of GTEx tissues powers discovery of disease and cell-type associated regulatory variants

Margaret K.R. Donovan[1,2], Agnieszka D'Antonio-Chronowska[3], Matteo D'Antonio [3]* & Kelly A. Frazer [3,4]*

The Genotype-Tissue Expression (GTEx) resource has provided insights into the regulatory impact of genetic variation on gene expression across human tissues; however, thus far has not considered how variation acts at the resolution of the different cell types. Here, using gene expression signatures obtained from mouse cell types, we deconvolute bulk RNA-seq samples from 28 GTEx tissues to quantify cellular composition, which reveals striking heterogeneity across these samples. Conducting eQTL analyses for GTEx liver and skin samples using cell composition estimates as interaction terms, we identify thousands of genetic associations that are cell-type-associated. The skin cell-type associated eQTLs colocalize with skin diseases, indicating that variants which influence gene expression in distinct skin cell types play important roles in traits and disease. Our study provides a framework to estimate the cellular composition of GTEx tissues enabling the functional characterization of human genetic variation that impacts gene expression in cell-type-specific manners.

[1] Bioinformatics and Systems Biology Graduate Program, University of California, San Diego, La Jolla, CA 92093, USA. [2] Department of Biomedical Informatics, University of California, San Diego, La Jolla, CA 92093, USA. [3] Department of Pediatrics and Rady Children's Hospital, University of California, San Diego, La Jolla, CA 92093, USA. [4] Institute for Genomic Medicine, University of California, San Diego, La Jolla, CA 92093, USA. *email: madantonio@health.ucsd.edu; kafrazer@health.ucsd.edu

Understanding the regulatory impact of genetic variation on complex traits and disease has been a longstanding goal of the field of human genetics. To decipher the mechanistic underpinnings of complex traits, the genotype-tissue expression (GTEx) project[1] has generated a large dataset, including over 10,000 bulk RNA-seq samples representing 53 different tissues (corresponding to 30 organs) obtained from 635 genotyped individuals, to link the influence of genetic variants on gene expression levels through quantitative trait loci analysis (eQTL). While GTEx provides important biological insights, unaccounted for cellular heterogeneity (i.e., different cell types within a tissue and the relative proportions of each cell type across samples of the same tissue) present in bulk RNA-seq can affect genotype–gene expression associations[2]. Since regulation of gene expression varies across cell types, not accounting for cellular composition could result in loss or distortion of signal from relatively rare cell types, thus characterization of cellular heterogeneity across all GTEx tissues is critical for more comprehensive eQTL studies. It is possible that future studies pursuing cell-type-associated eQTLs may utilize single cell approaches (e.g. single cell RNA-seq; scRNA-seq); however, non-trivial technical challenges, such as hard to dissociate tissues and low capture efficiencies, make the generation of a GTEx-scale single-cell expression dataset a substantial undertaking, which would take years to complete. Thus, as single-cell large-scale scRNA-seq collections progress, our present knowledge of how genetic variation influences cell-type-associated gene expression would greatly benefit from conducting eQTL analyses on bulk GTEx tissue samples whose cellular heterogeneity has been characterized through existing deconvolution methods[3–5].

To characterize the heterogeneity of bulk RNA-seq samples, gene signatures from cell types known to be present in a given tissue can be used to deconvolute the cellular composition (i.e. the proportion of each cell type). The signature genes needed to deconvolute a heterogeneous tissue can be obtained by analyzing scRNA-seq generated from an analogous tissue. However, there are relatively few human scRNA-seq resources currently available[6–10], and thus only a small fraction of GTEx tissues could be deconvoluted using gene expression signatures derived from existing human single-cell data. While human single-cell data is limited, the Tabula Muris exists[11], which is a powerful resource of scRNA-seq data from mouse including more than 100,000 cells from 20 tissue types (referred in the Tabula Muris resource as organs and tissues). A recent study showed that similar cell types in humans and mice share sufficient gene expression signatures to integrate scRNA-seq data between the two species[12], raising the possibility of utilizing the available scRNA-seq from mouse to generate the gene expression signatures for deconvolution of GTEx tissues.

To examine the feasibility of using mouse-derived gene expression signatures to deconvolute human tissues, we compare cellular composition estimates of GTEx liver and GTEx skin samples generated using human scRNA-seq to those generated using the Tabula Muris scRNA-seq resource. We show that the human and mouse single-cell data capture many overlapping cell populations and that using either human-derived or mouse-derived gene signatures to deconvolute the 175 GTEx liver samples and the 860 GTEx skin samples resulted in highly correlated estimated cellular compositions. We show that the main differences between the cell types identified using the human-derived versus mouse-derived signature genes are due to: (1) subtle biological differences that exist in human and mouse immune cells, and (2) resolution (i.e., the ability to detect less abundant cell types and distinguish between similar cell types) which was impacted by technical differences in the human and mouse scRNA-seq data

sets, including the number of cells captured and subjected to scRNA-seq and the spatial location from which the tissue was sampled. We use gene signatures derived from the Tabula Muris resource to deconvolute 6829 GTEx samples corresponding to 28 tissues from 14 organs, which enables us to determine how the fractions of different cell types vary across GTEx samples derived from the same tissue. Using deconvoluted liver and skin GTEx samples for eQTL analyses, we identify thousands of previously undetected genetic associations (i.e. not detected using bulk RNA-seq samples) that tend to have lower effect sizes, some of which are cell-type-associated. Finally, we show that skin cell-type-associated eQTLs colocalize with GWAS variants for melanoma, malignant neoplasm, and infection signatures, indicating that variants that are functional in limited skin cell types may play major roles in skin traits and disease. Taken together, our study demonstrates two major principles: (1) mouse-derived signature genes can be used to deconvolute the cellular composition of human tissues; and (2) the estimation of cellular heterogeneity by deconvolution enhances the genetic insights yielded from the GTEx resource.

## Results

**Mouse and human scRNA-seq capture similar cell types**. To examine the extent to which scRNA-seq generated from analogous human and mouse tissues (Supplementary Table 1) captured similar cell types, we first examined liver as a proof-of-concept tissue (Fig. 1a, *proof-of-concept*). We used previously defined cell types from Tabula Muris mouse liver cells (which were purified for viable hepatocyte and non-parenchymal cells followed by FACS sorting; 710 cells; 5 cell types)[11], and to be consistent, we used the Tabula Muris annotation approach to analyze existing human liver scRNA-seq data (total liver homogenate; 8119 cells; 15 cell types)[6]. In brief, on the 8119 human liver single-cells, we performed nearest-neighbor graph-based clustering on components computed from principal component analysis (PCA) of variably expressed genes, and then used marker genes to define the cell populations corresponding to each of the 15 previously observed cell types[6] (see the "Methods" section).

Human and mouse scRNA-seq from liver captured several shared cell types, including hepatocytes, endothelial cells, and various immune cells (Kuppfer cells, B cells, and natural killer (NK) cells) (Fig. 1b–e), however we noted that there were many more distinct cell types for human liver. This was due to the fact that cell type resolution (i.e. the ability to distinguish between similar cell types) can be influenced by: (1) the number of cells captured and subjected to scRNA-seq, which may influence the proportion of observed common or rare cell types[13]; and (2) how the tissue was sampled, which may enrich for selected populations or capture how populations are distinguished by spatial location (i.e. zonation). Some of the 15 cell types identified in the human liver scRNA-seq were highly similar and clustered near each other, for example there were four hepatocytes populations and two endothelial cell populations (human periportal sinusoidal endothelial cells (SEC) and central venous SECs) distinguished by their zonation (Fig. 1b, c). In contrast, for the mouse liver scRNA-seq, which had considerably fewer cells analyzed, we only observed one hepatocyte population and one endothelial population (Fig. 1d, e). If we collapsed the cell types that were similar to each other in the human scRNA-seq, we obtained seven distinct cell classes (Fig. 1b, f; Supplementary Table 2), which largely corresponded to the five cell types from mouse liver scRNA-seq (cholangiocytes and hepatic stellate cells were absent due to having been sorted by FACS; Fig. 1d–f). Overall, these results

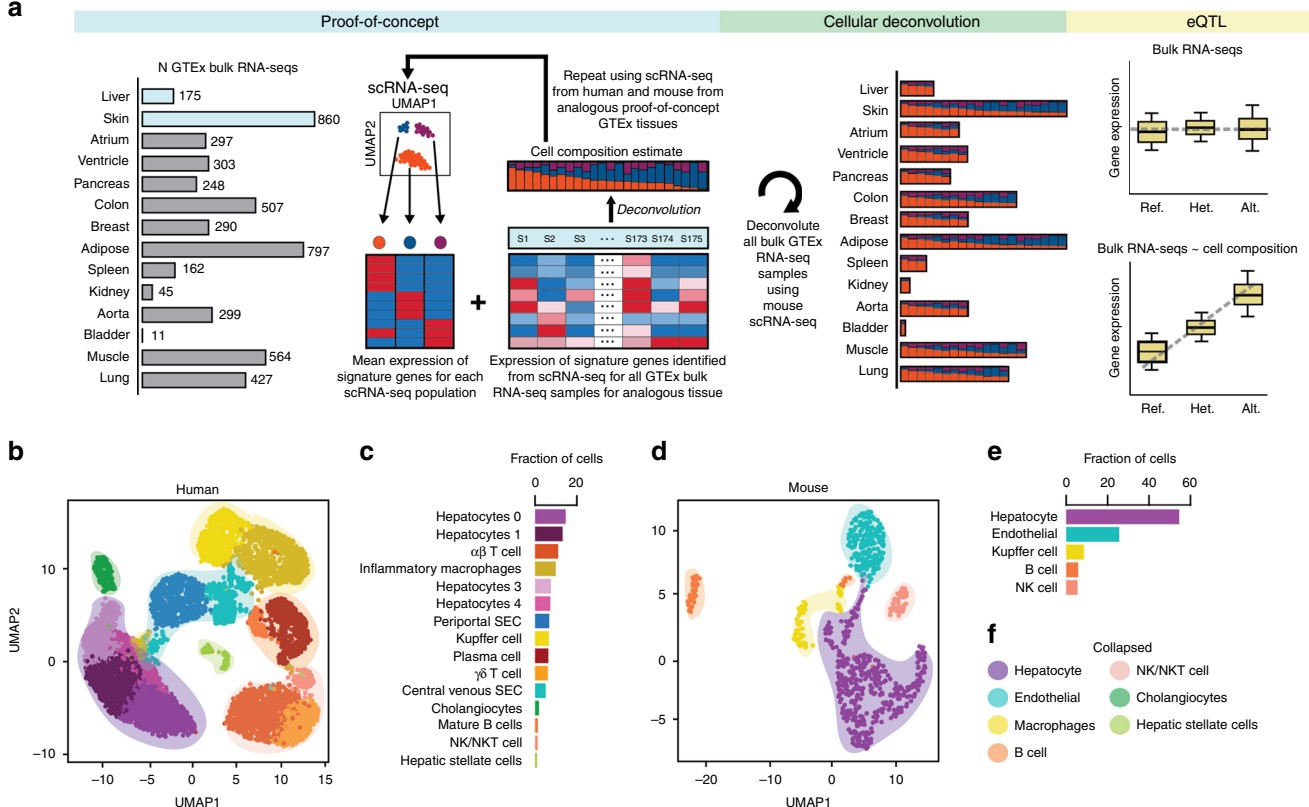

**Fig. 1 Human and mouse liver scRNA-seq contains similar cell types. a** Overview of the study design, which was to deconvolute the cellular composition of 28 GTEx tissues from 14 organs using mouse scRNA-seq to identify cell-type-associated eQTLs. We first conducted *proof-of-concept* analyses, where we compared cellular estimates of two proof-of-concept GTEx tissues (liver and skin) deconvoluted using both mouse and human signature genes obtained from scRNA-seq. We then performed *cellular deconvolution* of the 28 GTEx tissues from 14 organs using CIBERSORT and characterized both the heterogeneity in cellular composition between tissues and the heterogeneity in relative distributions of cell populations between RNA-seq samples from a given tissue. Finally, we used the cell type composition estimates as interaction terms for *eQTL analyses* to determine if we could detect cell-type-associated genetic associations. **b** UMAP plot of clustered scRNA-seq data from human liver. Each point represents a single cell and color coding of cell type populations are shown adjacent **c**. Similar cell types can be collapsed to single cell type classifications and are noted with colored, transparent shading **f**. **c** Bar plots showing the fraction of each cell type from human liver scRNA-seq data. Color-coding of cell types correspond to the colors of the single cells in **b**. **d** UMAP plot of clustered scRNA-seq data from mouse liver. Each point represents a single cell and color coding of cell type populations are shown adjacent **e**. Each cell type has a corresponding collapsed cell type in human liver and is noted with colored, transparent shading **f**. **e** Bar plots showing the fraction of each cell type from mouse liver scRNA-seq data. Color-coding of cell types correspond to the colors of the single cells in **d**. **f** showing the colors of collapsed similar cell types from human liver (transparent shading in UMAP **b**, **d**; Supplementary Table 2). All cell types from mouse liver have a corresponding collapsed cell type in human liver (hepatocyte, endothelial, macrophages, B cell, NK/NKT cell) and human liver also contains two additional cell types not present in mouse (cholangiocytes and hepatic stellate cells).

show that scRNA-seq generated from human and mouse liver captured similar cell types and that technical differences, including the number of cells analyzed and tissue sampling methodology, affects the cell type resolution.

**Mouse liver signature genes can deconvolute GTEx human liver.** To establish the ability to use expression profiles of signature genes derived from mouse scRNA-seq for the deconvolution of human GTEx tissues, we first examined if the similarly annotated cell types identified in the two species (Fig. 1b–e) clustered together based on their gene expression profiles. We harmonized the human and mouse liver scRNA-seq using canonical correlation analysis (CCA) and visualized using uniform manifold approximation and projection (UMAP) (Fig. 2a, b). We observed that the corresponding cell types across the two species clustered closely together, indicating that they had highly similar gene expression profiles.

We next compared the cellular composition estimates of 175 GTEx bulk liver RNA-seq samples[1] obtained by deconvolution

using human signature genes to those obtained using mouse signature genes (Fig. 1a, *proof-of-concept*), which respectively, consisted of the top 200 most significantly overexpressed genes for each cell type identified in scRNA-seq from high-resolution human liver (i.e. signature genes from 15 cell types) and low-resolution mouse liver (i.e. signature genes from 5 cell types) (Supplementary Data 1). From the 175 GTEx bulk liver RNA-seq samples, we independently extracted the expression of the signature genes at the two resolutions, and used CIBERSORT[3] to estimate the cellular compositions (i.e. high-resolution human liver estimates and low-resolution mouse liver estimates) (Fig. 2c, d; Supplementary Data 9, 10). To investigate how resolution impacted the correlation between human and mouse signature gene estimates, we also collapsed the high-resolution human liver cellular composition estimates for each of the 175 deconvoluted samples by summing the estimates across similar cell types in each of the 7 distinct cell classes (Supplementary Table 2) (Figs. 1b, f and 2e). We then calculated all pairwise-correlations between each of the estimated cell populations in the 175 GTEx

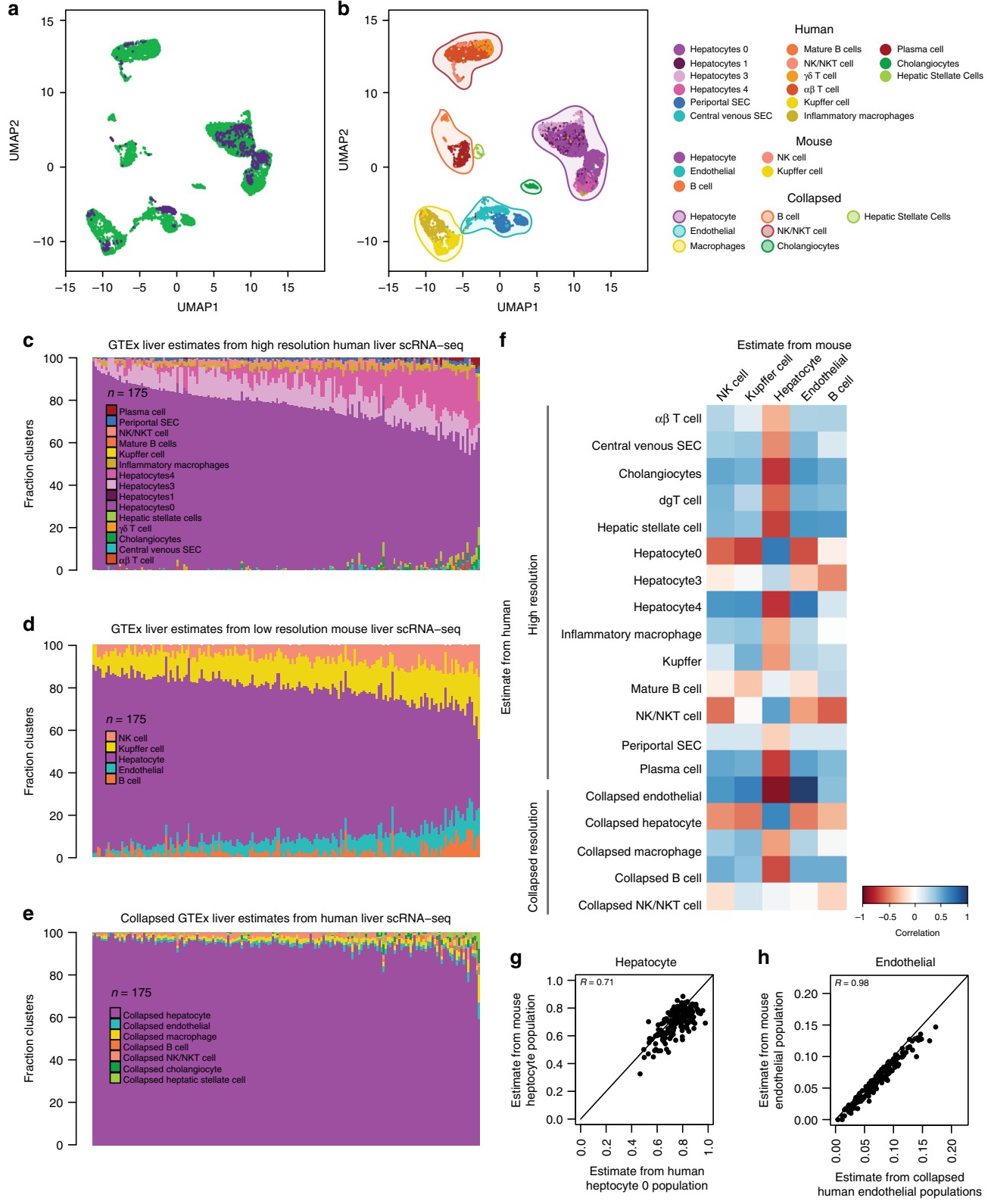

liver samples from human (high and collapsed resolution estimates) with the estimated cell populations from mouse (low-resolution estimates) (Fig. 2f, Supplementary Fig. 1). We found that hepatocyte estimates from mouse liver were positively and highly correlated with the human high-resolution hepatocyte 0 population estimate ($r = 0.71$, $p$-value $= 5.4 \times 10^{-28}$), but not correlated with any of the other three high-resolution hepatocyte

populations (1, 3, and 4); and was slightly less correlated with the collapsed hepatocyte population estimate ($r = 0.64$, $p$-value $= 1.015 \times 10^{-21}$) (Fig. 2f, g). This indicates that the low-resolution mouse hepatocyte population corresponds to one of the four human hepatocyte populations/zones potentially due to tissue sampling. Further, we observed that the endothelial estimates from mouse were highly correlated with the collapsed human

**Fig. 2 Comparison of GTEx liver cell estimates using mouse versus human signature gene. a** UMAP plot of integrated scRNA-seq data from human and mouse liver. Each point represents a single cell and color coding of cells indicates the species the cells were obtained from (human = green; mouse = purple). **b** UMAP plot of integrated scRNA-seq data from human and mouse liver. Each point represents a single cell and color coding of cell type populations are shown in the adjacent legend. The collapsed populations are the same as those shown in Fig. 1f. **c–e** Bar plots showing the fraction of cell types estimated in the 175 GTEx liver RNA-seq samples deconvoluted using gene expression profiles from high-resolution human liver scRNA-seq **c**, low-resolution mouse liver scRNA-seq **d**, and GTEx estimates generated by collapsing high-resolution human cell types within each of the seven distinct cell classes **e**. **f** Heatmap showing the correlation of GTEx liver cell population estimates from human liver scRNA-seq at high and collapsed resolutions (rows) and mouse liver (columns) at low resolution. Color coding of heatmap scales from red, indicating negative correlation in estimates, to blue, indicating positive correlation in estimates. Most correlations were significant (p-values are reported in Supplementary Data 2A). **g, h** Scatter plots of estimated cell compositions across 175 GTEx livers deconvoluted using human scRNA-seq for human hepatocyte 0 population **d** and human collapsed endothelial cells **e** versus estimated cell populations deconvoluted using mouse scRNA-seq.

endothelial population estimates ($r = 0.98$, $p$-value $= 1.2 \times 10^{-115}$) but not correlated with either high-resolution human periportal SECs or central venous SECs (Fig. 2h). This indicates that the human endothelial population estimates captured a higher resolution of cell type specificity (i.e. two independent endothelial zones), whereas the mouse endothelial population estimates likely captured a mixture of both cell types (i.e. the two endothelial zones are combined into a single cell population), which is potentially due to the lower number of mouse cells analyzed. While in general we observed high correlation in the human and mouse population estimates for most cell types (hepatocytes, endothelial cells, and Kupffer cells), B cells were non-significantly correlated, and NK-like cells were negatively correlated (Fig. 2f, Supplementary Data 2A). This difference in immune cell estimates in GTEx liver is not wholly unexpected, as biological differences, including immune response differences, exist between species[14]. To further examine the accuracy of the deconvolution, we conducted simulations to obtain 100 human liver samples with known cell type distributions, and confirmed that the estimated cell population distributions obtained using both human and mouse gene expression signatures were consistent with their expected values (Supplementary Fig. 2, Supplementary Data 3). Our results show that, while technical differences in scRNA-seq generation and biological differences between humans and mice may impact cell estimation performance, overall mouse signature genes can be used to deconvolute human GTEx bulk RNA-seq samples.

**Mouse skin signature genes can deconvolute GTEx human skin**. To examine the similarity of cellular estimates across 860 GTEx human skin samples obtained using human-derived versus mouse-derived signature genes, we used scRNA-seq from human epidermal cells[15] (digested dorsal forearm skin biopsies; 5670 cells; 9 cell types) (Fig. 3a, b) and Tabula Muris mouse skin cells (FACS sorted epidermal keratinocytes; 2263 cells; 6 cell types) (Fig. 3c, d). While the previous human and mouse liver scRNA-seq studies[6,11] used similar naming conventions for the cell type annotations (Fig. 1b–e), the human and mouse skin scRNA-seq studies[11,15] did not (Fig. 3a–d), and thus we first needed to identify the corresponding cell types across the two species. To accomplish this, we harmonized the human dermis and mouse skin scRNA-seq using CCA (Fig. 3e, f) and visualized using UMAP. We observed three distinct superpopulations: (1) superpopulation 1, epidermal cells, consisting of the four human keratinocyte populations (14, 5, 711, and 1) and mouse epidermal cells, basal cells, stem cells of epidermis, and outer bulge cells (keratinocyte stem cells), (2) superpopulation 2, consisting of the three human fibroblast populations (0, 3, and 4) and mouse inner bulge cells (keratinocyte stem cells); and (3) superpopulation 3, leukocytes, consisting of human lymphocytes and mouse leukocytes (Fig. 3e, f). Further, the different cell types within each of

the three clusters expressed corresponding marker genes (Supplementary Fig. 3c, f), confirming that they indeed were similar cell types in the human and mouse skin scRNA-seq. Overall, we found human and mouse skin scRNA-seq captured shared cell types that cluster into three distinct superpopulations.

We next compared the cellular composition estimates of 860 GTEx bulk skin RNA-seq samples[1] obtained by deconvolution using human gene expression signatures to those obtained by using mouse gene expression signatures (Fig. 1a, *proof-of-concept*). We obtained signature genes for each cell type identified in scRNA-seq from human skin (i.e. signature genes from each of 9 dermis cell types) and mouse skin (i.e. signature genes from each of 6 skin cell types) (Supplementary Data 1) and used CIBERSORT to deconvolute the 860 GTEx skin RNA-seqs (Fig. 3g, h). Given the presence of the three superpopulation clusters observed in the mouse and human scRNA-seq integration analysis (Fig. 3f), similar to liver, we investigated how resolution impacted the correlation between human and mouse signature gene estimates. We independently collapsed the high-resolution human epidermis (9 cell types) and the high-resolution mouse skin (6 cell types), by summing the estimates across the cell types in each of the three distinct superpopulations (Supplementary Table 3). We then calculated all pairwise-correlations between each of the estimated cell populations in the 860 GTEx skin samples from human estimates (high and collapsed) with the estimated cell populations from mouse (high and collapsed resolution) (Fig. 3i, Supplementary Fig. 4). Using the integration analysis (Fig. 3f) as a guide, we examined the similarity of estimates from human and mouse cell populations mapping to each of the three superpopulations. First, we examined the similarity of human cell types in Superpopulation 1 (Keratinocyte 14, Keratinocyte 5, Keratinocyte 711, Keratinocyte 1, cornified envelope, and collapsed estimates of these cell types) and mouse cell types in Superpopulation 1 (epidermal cell, basal cell, stem cell of epidermis, outer bulge, and collapsed estimates of these cell types) (Fig. 3f; dark purple shading). We observed the human keratinocyte population 14 had a strong positive correlation with the mouse stem cell of the epidermis estimates ($R = 0.89$; $p = 2.4 \times 10^{-103}$) (Fig. 3i, j). We also found that collapsed mouse epidermal cell estimates were correlated with collapsed human keratinocyte population estimates ($R = 0.44$, $p = 1.19 \times 10^{-43}$) (Fig. 3i, k). These results indicate that despite differences in annotations, estimates from mouse and human cell types mapping to the epidermal cell superpopulation are highly correlated. Second, we examined the similarity of human cell types in superpopulation 2 (fibroblast 0, fibroblast 3, fibroblast 4, and collapsed estimates of these cell types) and the single mouse cell type (inner bulge) in this cluster (Fig. 3f; light purple shading). We found that human fibroblast (high resolution and collapsed) estimates were not correlated with the mouse inner bulge cell population estimates (Fig. 3i), indicating that, despite similar enough global gene expression patterns for the

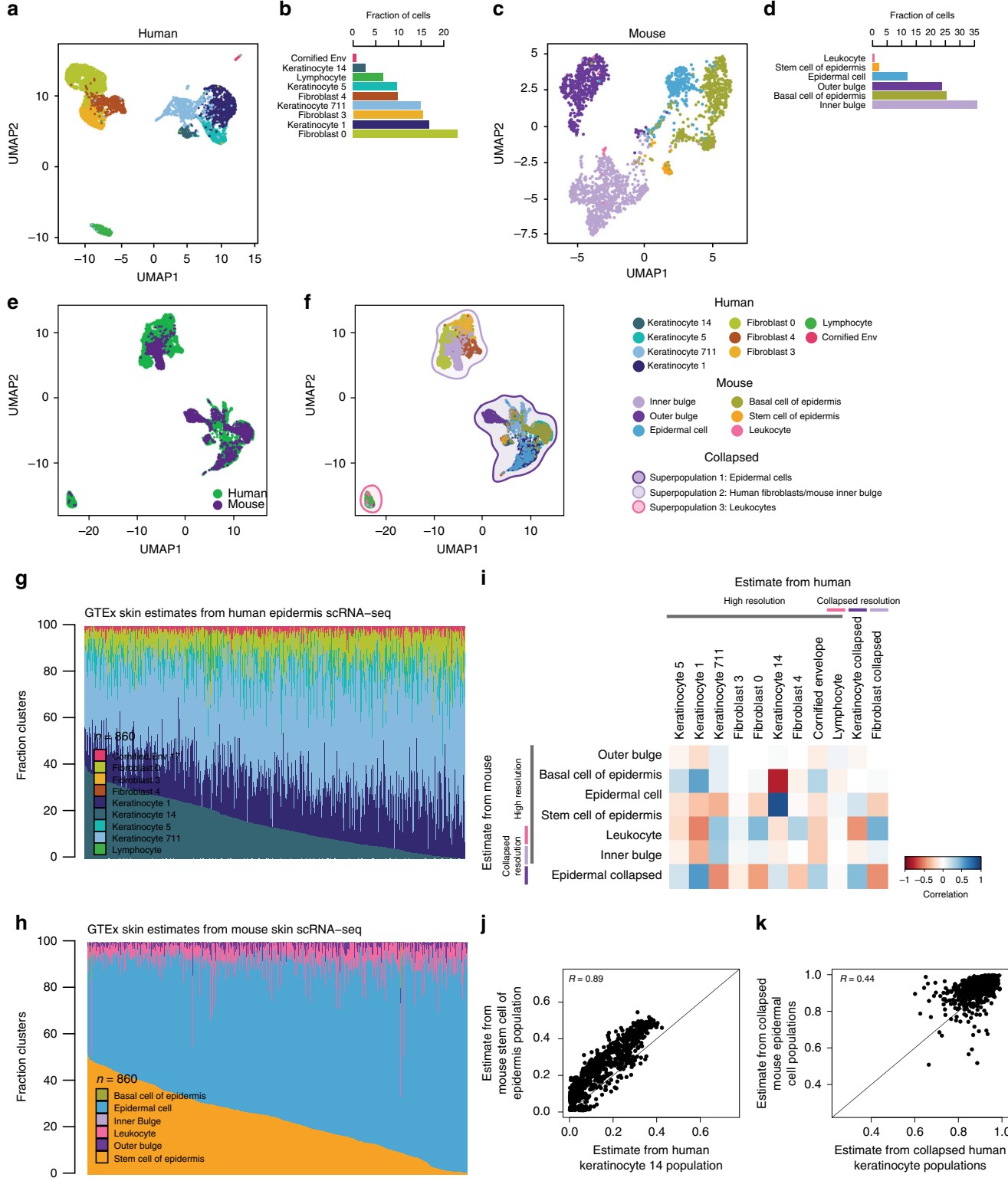

human fibroblasts and mouse inner bulge cells to cluster together, their signature genes distinguish them as different cell types during deconvolution. Third, we examined the similarity of the human cell type (lymphocyte) and mouse cell type (leukocyte) in superpopulation 3 (Fig. 3f; pink shading). Similar to the liver estimates, mouse and human leukocyte estimates were not correlated (Fig. 3i, Supplementary Data 2B), likely due to known species differences in immune cells. As we observed in liver, we confirmed that technical and biological differences influence cell

estimate performance, however overall cell composition estimates derived from human and mouse skin signature genes are correlated, supporting our ability to use mouse scRNA-seq as an alternative to human scRNA-seq for the deconvolution of GTEx tissues.

**Deconvolution of GTEx tissues reveals striking heterogeneity.**
To understand the extent to which the mouse signature genes obtained from cell types across 14 tissues were able to distinguish

**Fig. 3 Comparison of GTEx skin cell estimates using mouse versus human signature genes. a** UMAP plot of clustered scRNA-seq data from human epidermis. Each point represents a single cell and color coding of cell type populations are shown adjacent **b**. **b** Bar plots showing the fraction of each cell type from the scRNA-seq data from human epidermis. Color-coding of cell types correspond to the colors of the single cells in **a**. **c** UMAP plot of clustered scRNA-seq data from mouse skin. Each point represents a single cell and color coding of cell type populations are shown adjacent in **d**. **d** Bar plots showing the fraction of each cell type from the scRNA-seq data from mouse skin. Color-coding of cell types correspond to the colors of the single cells in **c**. **e** UMAP plot of integrated scRNA-seq data from human epidermis and mouse skin. Each point represents a single cell and color coding of cells indicates the species the cells were obtained from (human = green; mouse = purple). **f** UMAP plot of integrated scRNA-seq data from human epidermis and mouse skin. Each point represents a single cell and color coding of cell type populations and collapsed superpopulations are shown in the adjacent legend. **g, h** Bar plots showing the fraction of cell types estimated in GTEx skin RNA-seq samples from human epidermis scRNA-seq **g** and mouse skin scRNA-seq **h**. **i** Heatmap showing the correlation of GTEx skin cell population estimates from mouse skin scRNA-seq at high and collapsed resolutions (rows) and human skin (columns). Color coding of heatmap scales from red, indicating negative and low correlation in estimates, to blue, indicating positive and high correlation in estimates. Most correlations were significant (p-values are reported in Supplementary Data 2B). **j, k** Scatter plots of estimated cell compositions across 860 GTEx skin samples deconvoluted using human scRNA-seq for human keratinocyte 14 population versus mouse stem cell of epidermis population **j** and keratinocyte 1, 5, 14, 711 population versus collapsed mouse epidermal cell populations **k**.

between the 28 GTEx tissues, we extracted the expression of the signature genes (Supplementary Table 1, Supplementary Data 1) across the 6829 bulk GTEx RNA-seq samples and visualized how the samples clustered (Fig. 4a). We observed that the mouse signature genes were able to differentiate between the human GTEx organs, as well as illustrated the existence of organ sub-structures delineating heterogeneity in tissues belonging to the same organ. For example, tissues from the same organ clustered closely together and distinctly from other organs, including the heart tissues (atrium and ventricle), brain tissues (cortex, frontal cortex, hippocampus, anterior cingulate cortex, amygdala, sub-stantia nigra, spinal cord, putamen, nucleus accumbens, caudate, and hypothalamus), adipose (visceral and subcutaneous), and colon (sigmoid and transverse). Of note, within the brain we also observe clustering according to zonation, including clustering of samples from the cerebellum and cerebellar hemisphere, as well as clustering of samples from the frontal cortex, cortex, and anterior cingulate cortex. These results suggest that signature genes capture both expression differences between organs, as well as differences between cell types within tissues belonging to the same organ.

To understand the cellular heterogeneity of 28 GTEx tissues (Fig. 1a, *cellular deconvolution*), we used the signature genes from 14 mouse tissue types (Supplementary Table 1, Supplementary Data 1) to perform cellular deconvolution of 28 GTEx tissues from 14 organs (Fig. 4b; Supplementary Tables 1, 4, 5; Supplementary Data 4–17), where the number of samples for each GTEx tissue varied from 11 (bladder) to 860 (skin). We found that all samples were well-deconvoluted (p-value < 0.001; CIBERSORT, 1000 permutations) and that each deconvoluted GTEx tissue contained a variable number of cell types ranging from two (bladder) to seven (brain and heart) (Fig. 4c). In ~30% of the tissues (9 out of 28), we found that not all mouse cell types were estimated, possibly due to the GTEx tissues having been isolated for bulk RNA-seq from a different spatial location than mouse or species differences in cell types. Additionally, the relative distribution of the estimated cell types varied between different samples of the same tissue (Fig. 4d). Tissues with the least heterogeneous cell population distributions between samples were aorta and spleen (Supplementary Data 4,16), whereas those with the most heterogeneous cell population distributions between samples were brain (13 tissues), colon, and left ventricle (Supplementary Data 6,7,17). Examining the tissues correspond-ing to the same organ, we noted that some had the same cell types estimated at similar distributions (adipose subcutaneous and visceral), some had the same cell types present at variable proportions (heart atrial appendage and left ventricle; 13 brain tissues), and others had variable cell types present/absent (colon transverse and sigmoid). These results reveal a striking

heterogeneity in GTEx tissues that has not been previously appreciated and may be contributing noise to eQTL analyses.

**eQTL analyses using deconvoluted tissues increases power.** Since we observed heterogeneity in the relative distributions of cell populations across GTEx RNA-seq samples, we hypothesized that considering the cell population distributions of each sample would improve eQTL analysis by increasing our power to detect tissue and/or cell type associations (Fig. 1a). We identified 19,621 expressed genes in GTEx liver samples and performed one eQTL analysis not considering cellular heterogeneity (i.e. bulk resolu-tion; Supplementary Data 18), and three eQTL analysis using cell population estimates as covariates to adjust for cellular hetero-geneity (Supplementary Data 19–21): (1) considering high-resolution human liver estimates (15 cell types; Supplementary Data 9, 19; Fig. 2a); (2) considering collapsed resolution human liver estimates (7 cell types; Supplementary Table 2; Supple-mentary Data 9, 20; Fig. 2c); and (3) considering low-resolution mouse liver estimates (5 cell types; Supplementary Data 10, 21; Fig. 2b). Using cell population estimates as covariates we detected many more genes with significant eQTLs (eGenes) than at bulk resolution (Fig. 5a). We found that considering high-resolution estimates identified the most eGenes (10,117) with 1.3 fold and 3.1 fold more than collapsed and low-resolution estimates, respectively. These findings show that conducting eQTL analyses using highly resolved cell population estimates as a covariate significantly increases the power to identify eGenes.

Given the differences in the number of detected eGenes based on cell-type resolution, we hypothesized that eGenes detected at low powered resolutions (bulk and collapsed resolution) com-monly shared eQTLs with other GTEx tissues (i.e. tissue-neutral) and the eGenes detected using higher powered resolutions had more tissue-associated eQTLs (i.e. less frequently in other GTEx tissues). For each resolution, we calculated the number of GTEx tissues in which each eGene has eQTLs. We observed that eGenes identified using cell populations as covariates in general were more tissue-associated than eGenes detected at bulk resolution. Compared to bulk resolution, high resolution eGenes were the most tissue-associated ($p = 4.2 \times 10^{-194}$; Mann–Whitney $U$ test), then low-resolution eGenes ($p = 2.17 \times 10^{-174}$; Mann–Whitney $U$ test), and collapsed resolution was the least tissue-specific ($p = 6.59 \times 10^{-94}$; Mann–Whitney $U$ test) (Fig. 5b), showing that the resolution of cell population estimates used as covariates is correlated with the power of the study to identify tissue-associated eGenes.

Furthermore, using cell populations as covariates resulted in decreased effect size ($\beta$) (Fig. 5c) and standard error (SE) of $\beta$ (Fig. 5d), where relative to bulk resolution, the higher the

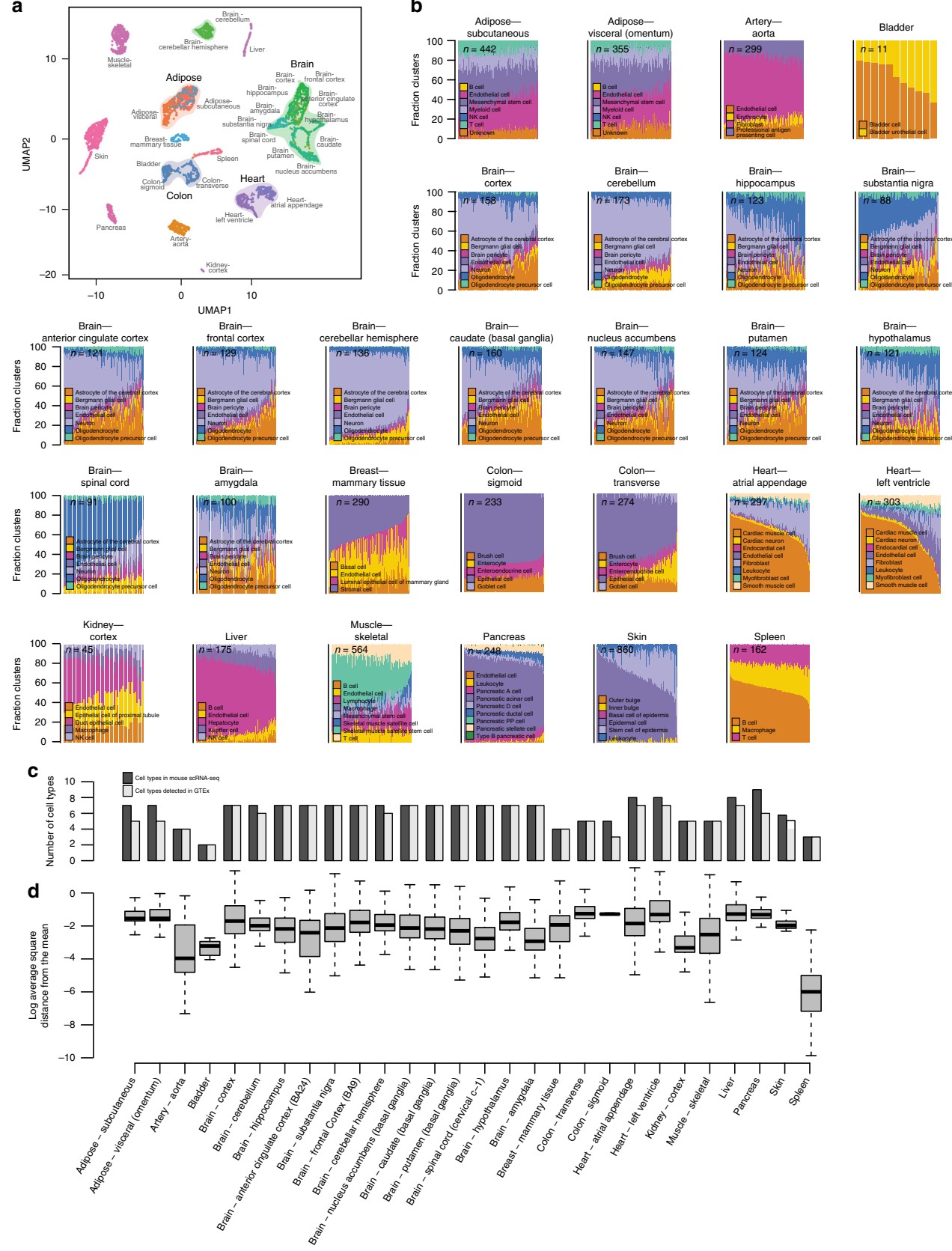

**Fig. 4 Cellular deconvolution of 28 GTEx tissues. a** UMAP using the expression of all scRNA-seq-derived signature genes across the 28 GTEx tissues. **b** Stacked bar plots showing the fraction of cell types estimated in GTEx RNA-seq samples from mouse scRNA-seq. A colorblind-friendly version of this figure is shown in Supplementary Fig. 5. **c** Bar plots comparing the number of cell types discovered in mouse scRNA-seq (light gray) vs. the number of these cell types that were estimable for each GTEx tissue. **d** Box plots showing per RNA-seq sample the distribution of the $\log_2$ average square distance from the mean estimated cellular compositions for each GTEx tissue. The thick, black line indicates the median and the dashed lines indicate the bounds of the upper and lower whiskers.

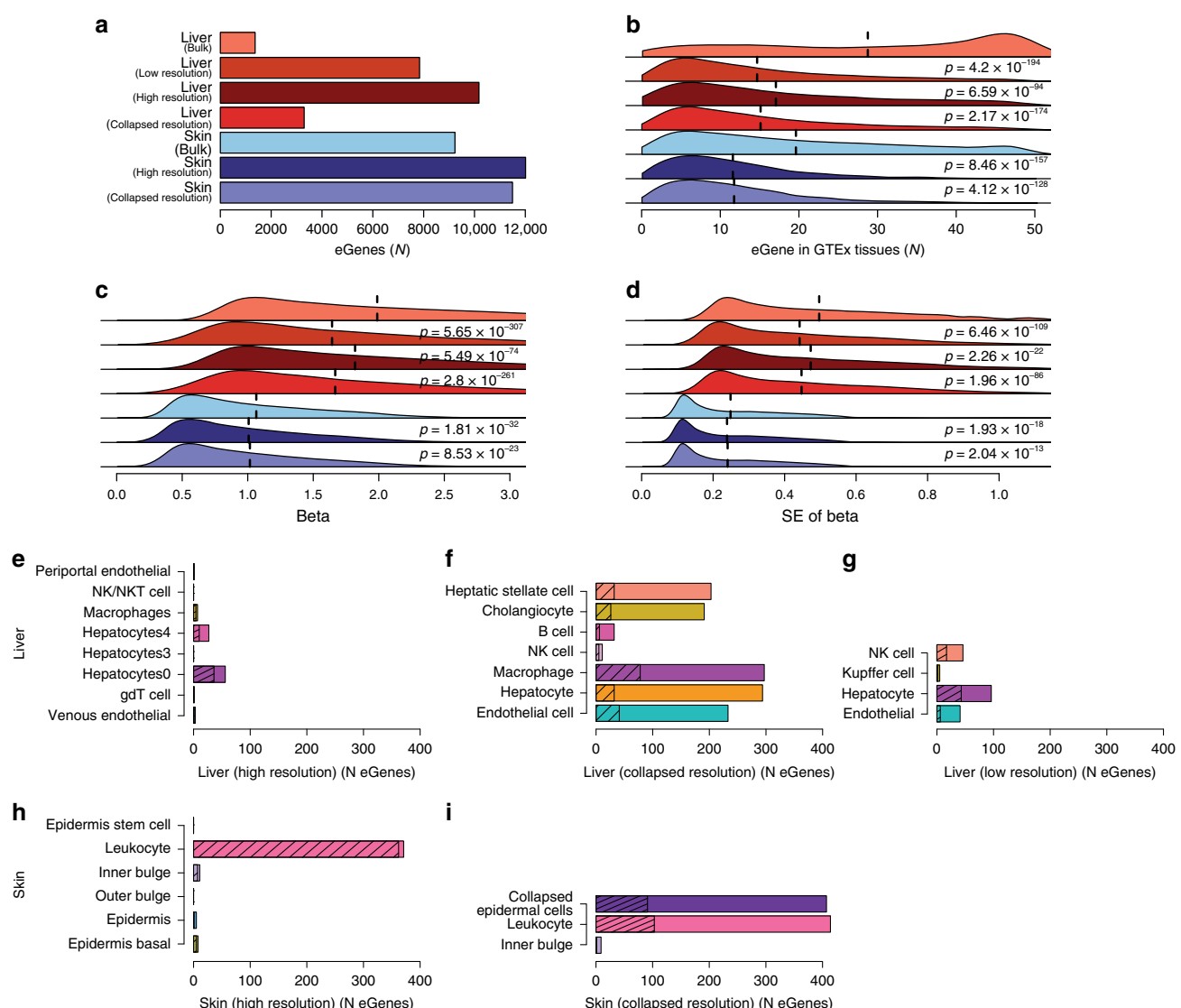

**Fig. 5 Using cellular deconvolution to discover cell-type-associated eQTLs. a** Bar plot showing the number of eGenes detected in each eQTL analysis from liver (shades of red) and skin (shades of blue). **b**-**d** Distributions of **b** number of GTEx tissues where each eGene has significant eQTLs, **c** effect size $\beta$, and **d** standard error of $\beta$ in liver and skin. Colors are as in panel **a**. Vertical dashed lines represent mean values. *p*-values were calculated in comparison with the bulk resolution analysis for each tissue using Mann–Whitney $U$ test. **e**-**i** Bar plots showing the number of eGenes significantly associated with each cell population considering cell estimates for: liver high resolution **e**, liver collapsed resolution **f**, liver low resolution **g**, skin high resolution **h**, and skin collapsed resolution **i**. Total number of eGenes for each cell type indicates the cell type is significantly associated and the hashed number of eGenes for each cell type indicates the association is cell-type-specific (e.g. only significant in that cell type). In cases where a given cell type had no significant association, the bar is not shown.

resolution of the eQTLs, the smaller the $\beta$ and SE of $\beta$. However, in general the $\beta$ values for the top hit for each gene were highly correlated between eQTLs detected using cell populations and eQTLs detected without using cell populations ($r > 0.975$, Supplementary Fig. 7A–C). By performing a permutation test, we confirmed that the detection of a larger number of eQTLs

using cell populations was biologically relevant, rather than simply resulting from the fact that a larger number of covariates were used (Supplementary Fig. 8). These results indicate that using cell population distributions as covariates overall reduces the noise, thereby potentially increasing our power to identify eQTLs.

**Resolution impacts cell-type-associated eQTL detection**. To examine if some of the eQTLs identified using cell population estimates as covariates were cell-type-associated, we used a statistical interaction test[16–18] to assess if modeling the contribution of a cell type significantly improved the observed association between genotype and gene expression. Interaction tests were performed on all independent pairs of eGenes and corresponding lead eQTLs using liver cell type estimates from the high, collapsed, and low resolution as interaction terms. Overall, across the high, low, and collapsed resolutions we, respectively detected 74, 121, and 528 cell-type-associated eGenes (i.e., eGene is more significant considering cell type estimates but associated with one or more cell type(s); FDR-corrected $p$-values < 0.1, $\chi^2$ test, Fig. 5e–g) and 54, 68, and 220 cell-type-specific eGenes (i.e. eGene is associated with only one cell type; Fig. 5e–g). We investigated if relative cell abundance influenced our ability to detect cell-type-associated eGenes (i.e., is there more power for high abundance cells) and determined that it did not play a factor (Supplementary Fig. 9). Further, we noted by using low resolution and collapsed resolution cell populations, we respectively detected 1.6 and 7.1 times more cell-type-associated eQTLs than high-resolution cell populations (respectively, $p = 1.9 \times 10^{-7}$ and $7.3 \times 10^{-250}$, Fisher's exact test, Fig. 5e–g). While initially counterintuitive to the previous evidence showing higher resolution eGenes are more tissue-specific (Fig. 5b) and have decreased noise (Fig. 5c, d), it is possible we identify a greater number of cell-type-associated eGenes using low resolution cell population estimates due to prevention of the dilution of eQTL signals between shared cell types, as might occur in cases where a regulatory variant has similar effects across similar cell types. Overall, these results suggest that accounting for cellular heterogeneity between samples allows for the discovery of cell-type-associated (and cell-type-specific) eQTLs.

**Detection of cell-type-associated eQTLs in GTEx skin**. To further investigate the impact of using cell populations on power to identify previously undetected eGenes and cell-type-associated eQTLs, we conducted eQTL analyses using the GTEx tissue (skin), which includes the largest number of RNA-seq samples (Fig. 4b). Although we deconvoluted 860 skin RNA-seqs using signature genes from high-resolution mouse skin scRNA-seq (6 cell types; Fig. 3b), only 749 had corresponding genotypes from 510 distinct individuals. We identified 24,029 expressed genes in the 749 skin RNA-seq samples with corresponding genotypes and performed three eQTL analyses: (1) without considering cell population distributions (bulk resolution) (Supplementary Data 22); (2) considering high-resolution mouse skin cell estimates (6 cell types; Supplementary Data 14, 23; Fig. 3c); and (3) considering collapsed resolution mouse skin cell estimates (3 cell types; Supplementary Data 2,14, 24; Fig. 3c, f). Using cell (high and collapsed) population distributions as covariates, respectively, we detected a 30% and 24% increase in eGenes with significant eQTLs (12,011 and 11,497 compared with 9232, Fig. 5a). Similar to our observation in liver, we found that eGenes specific for the eQTL analysis performed using high and collapsed cell populations as covariates, respectively, had eQTLs in fewer tissues than eGenes detected at bulk resolution ($p = 8.46 \times 10^{-157}$; $p = 4.12 \times 10^{-128}$, Mann–Whitney $U$ test; Fig. 5b), had a decreased effect size $\beta$ ($p = 1.81 \times 10^{-32}$; $p = 8.53 \times 10^{-23}$, Mann–Whitney $U$ test, Fig. 5c), and had decreased standard error (SE) of $\beta$ ($p = 1.93 \times 10^{-18}$; $p = 2.04 \times 10^{-13}$, Mann–Whitney $U$ test, Mann–Whitney $U$ test; Fig. 5d). We also observed that the $\beta$ values for the top hit for each eGene were highly correlated between eQTLs detected using high and collapsed cell populations and eQTLs detected without using cell populations ($r = 0.994$; $r = 0.996$, Supplementary Fig. 7d).

Further, at high resolution we detected 384 cell-type-associated eGenes (FDR-corrected $p$-values < 0.1, $\chi^2$ test, Fig. 5h) and 375 cell-type-specific eGenes (FDR-corrected $p$-values < 0.1, $\chi^2$ test, Fig. 5h), which were predominantly associated with leukocytes, while at collapsed resolution we detected 511 cell-type-associated eGenes (FDR-corrected $p$-values < 0.1, $\chi^2$ test, Fig. 5i) and 220 cell-type-specific eGenes (FDR-corrected $p$-values < 0.1, $\chi^2$ test, Fig. 5i), associated with both the collapsed epidermal cell population and leukocytes (Superpopulations 1 and 3; Fig. 3f). We hypothesize that substantially fewer cell-type-specific associations were observed in the high-resolution epidermal cell types (epidermal cell, basal cell, stem cell of epidermis, outer bulge; Fig. 5h) compared with the collapsed epidermal cells (Fig. 5i), because of a dilution of signal between similar cell types. The relatively large number of cell-type-associated eGenes in skin compared with the liver could be reflective of sample size differences between the two tissues (749 and 153, respectively) impacting power to detect eGenes. These results show that even in eQTL studies using large sample sizes, accounting for cellular heterogeneity results in the detection of thousands more eGenes, which tend to show cell-type-associated differential regulation.

**Skin cell-type-associated eQTLs colocalize with skin disease**. To explore the functional impact of the cell-type-associated eQTLs identified in skin, we examined their overlap with GWAS signals for skin traits and disease. From the UK Biobank, we extracted GWAS summary statistics for 23 skin traits where the cell types identified from skin scRNA-seq (Fig. 6a) likely played a role in the traits (Supplementary Data 25) and grouped them into seven categories based on trait similarity: (1) malignant neoplasms, (2) melanomas, (3) infections, (4) ulcers, (5) congenital defects, (6) cancer (broad definition, non-malignant neoplasm), and (7) unspecified skin conditions. As the three collapsed skin superpopulations identified the most cell-type-associated eGenes, we performed colocalization of the eQTLs identified using the collapsed resolution cell estimates (Supplementary Table 3) and skin GWAS loci to identify shared causal variants using coloc[19] and examining instances with PP4 > 0.5 (PP4, posterior probability of the colocalization model having one shared causal variant). We identified 394 variants that showed evidence of colocalization (Supplementary Data 25). These results show that we could identify hundreds of skin eQTLs that likely share a causal variant with skin GWAS traits.

We next asked if skin GWAS traits were enriched for eQTLs that are associated with distinct cell types. We tested the enrichment of cell-type-associated eQTLs at multiple PP4 thresholds and found malignant neoplasms and melanomas were enriched for eQTLs associated with keratinocyte stem cells from the inner bulge ($p = 1.13 \times 10^{-3}$, $p = 2.82 \times 10^{-4}$ Fisher's Test; Fig. 6b, c), and infections were enriched for eQTLs associated with leukocytes ($p = 9.69 \times 10^{-3}$ Fisher's Test; Fig. 6d). We did not observe a significant enrichment of cell-type-associated eQTLs in ulcers (Fig. 6e), congenital malformations (Fig. 6f), cancer (broad definition), or unspecified skin conditions. It is unclear if this is to be expected, as it is possible other cell types not estimated may be contributing to the diseases or in the case of congenital malformations, it is possible that expression differences impacting congenital malformations may be functioning during development and not detectable in adult skin. Overall, these results suggest that GWAS lead variants are commonly cell-type-associated regulatory variants, indicating that onset or progression of human disease and traits may be controlled at the cell type level.

We next sought to specifically examine the eGenes that most strongly colocalized with malignant neoplasms or melanoma

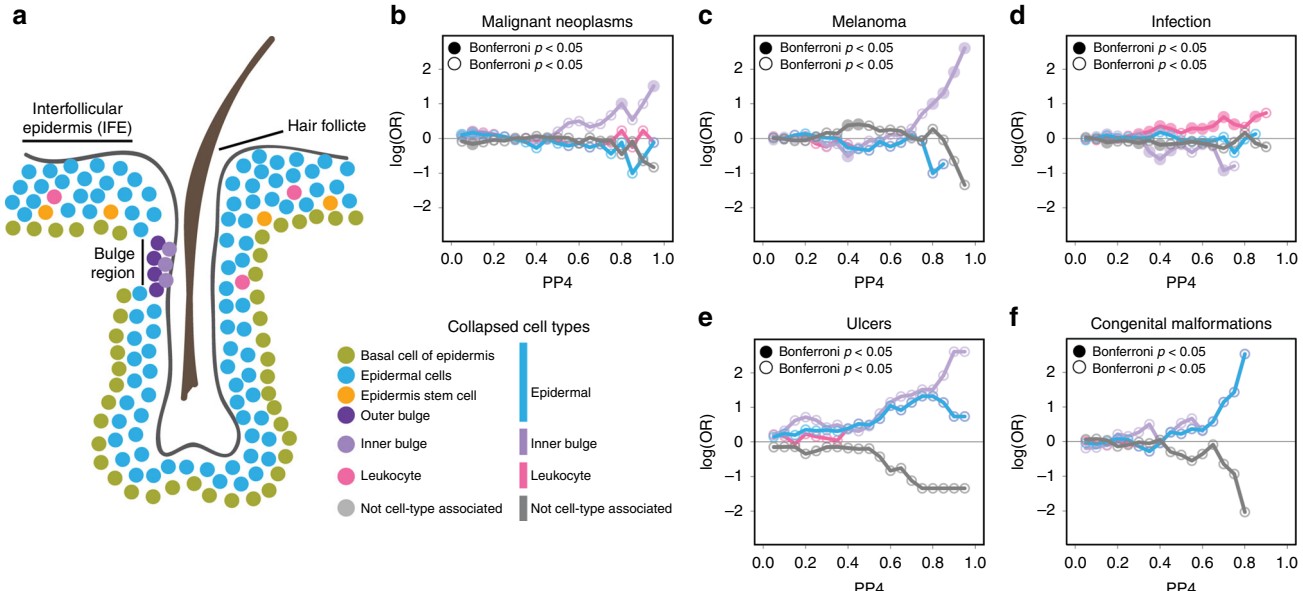

**Fig. 6 Colocalization of cell-type-associated skin eQTLs with skin GWAS traits. a** Cartoon describing the approximate organization of cell types identified in scRNA-seq from skin. Colors used for each cell type are used throughout figure and described in the adjacent legend. **b–f** Line plots showing the enrichment of cell-type-associated eQTLs in various GWAS traits: malignant neoplasms **c**, melanoma **d**, infection **e**, ulcers **f**, and congenital malformations **g**. Enrichment is plotted as the log(OR) (y-axis) over all probabilities of the eQTL signal overlapping (0 = not overlapping–1 = completely overlapping) with the GWAS signal (x-axis). Lines are colored following color coding of each cell type from Fig. 5a.

(PP4 ≥ 0.8), as bulge stem cells have been implicated in playing a role in cancer[20–25]. We found six eGenes not previously associated with skin cancers with eQTLs significantly associated with inner bulge stem cells, including: (1) *BRIX1*, which has been found to play a role in cancer progression[26]; (2) *RP11-875011*.1, an antisense gene, which has not previously been implicated in cancer, however antisense genes are thought to contribute to the regulation of human cancers[27]; (3) *MUL1*, which has been associated with the progression of human head and neck cancer[28]; (4) *PMS2P3*, has been implicated in affecting survival in pancreatic cancer[29]; (5) *FTH1*, which has been shown to be involved in regulating tumorigenesis[30,31] and whose increased expression in keratinocytes may be in response to stress[32,33]; and (6) *CNTN2*, which is involved in cell adhesion and has been implicated in tumor development[34,35]. The identification of these disease-associated eGenes supports our ability to identify skin cell-type-associated eQTLs whose functions are congruent with playing a role in the etiology of cancer. Together these results show that conducting eQTL studies accounting for cellular heterogeneity can identify the likely causal cell-type-associated variants and genes underlying GWAS disease loci.

## Discussion
Human scRNA-seq data representative of all tissues in GTEx that could be used to deconvolute the more than 10,000 GTEx bulk RNA-seq samples does not yet exist. As the Tabula Muris resource of mouse scRNA-seq from 20 organs was recently released[11], we sought to determine if mouse signature genes obtained from scRNA-seq could be used as an alternative for human signature genes for cellular deconvolution of GTEx RNA-seq samples. Using scRNA-seq from both mouse and human for two proof-of-concept tissues (liver and skin), we derived signature genes and used these expression profiles to deconvolute GTEx liver and skin RNA-seq samples. In general, human and mouse estimates between the two proof-of-concept tissues were comparable, where discrepancies in cell composition estimates between the two species primarily resulted from technical and subtle immunological differences. Specifically, in both liver and skin, technical differences impact the resolution at which cellular composition can be estimated, including: (1) the number of cells captured and subjected to scRNA-seq; and (2) tissue sampling methodology. Further, differences in cell composition estimates for immune cells were observed most likely due to immunological differences between the two species. These differences highlight that high-resolution scRNA-seq (more cells/cell types sampled from diverse zones) is key to identifying and estimating the composition of highly specialized and rare cell types. For these reasons, the cell composition estimates we obtained from CIBERSORT using mouse-derived signature genes from proof-of-concept liver and skin scRNA-seq may still be missing cell types not captured in the Tabula Muris resource. An additional challenge we found that influenced our ability to compare cell composition estimates was the scRNA-seq cell annotations in human and mouse skin did not use consistent naming conventions, thus it was not immediately clear how to compare cell estimates across the studies. We were able to overcome this challenge by integrating the mouse and human scRNA-seq, which allowed us to infer three similar superpopulations of cell types across the two species based on gene expression.

To examine cellular heterogeneity across the GTEx resource, we used the signature genes obtained using scRNA-seq from 14 mouse tissue types to deconvolute 6829 GTEx RNA-seq samples mapping to 28 tissues from 14 organs. We found that GTEx tissues exhibit substantial cellular heterogeneity, with the number of cell types ranging from two in bladder to seven in brain and heart. Additionally, some of the tissues, including brain, colon, and left ventricle, showed highly variable proportions of estimated cell types between samples, contributing to intra-tissue cellular heterogeneity. Together, these results reveal a source of heterogeneity in GTEx tissues that has not been previously considered and may contribute to reduced power to detect eQTLs.

While genetic association studies performed by GTEx have identified a wealth of insights into how human genetics function across bulk tissues[1], these analyses have not considered

how cellular heterogeneity can confound these studies through biasing or even masking cell-type-specific signals. We found that considering cellular heterogeneity significantly improved eQTL analyses by increasing power to detect lower effect size genetic associations, as well as by identifying cell-type-specific associations that were masked in analyses using bulk RNA-seq data from the same samples. Further, we found resolution of cellular heterogeneity influenced eQTL results, where considering high-resolution estimates identified substantially more eQTLs than using lower resolutions (low resolution or collapsed resolution); however, high-resolution cell estimates identified fewer cell-type-associated genetic associations than lower resolutions. It is possible this decrease in associations may be due to a dilution of signal between similar cell types. Our observations suggest these two resolutions should both be used to power eQTL analyses in complementary ways: (1) high-resolution estimates to power association analyses to discover lower effect size eQTLs; and (2) collapsed resolution estimates to identify cell-type associated eQTLs. We further show that cell-type-associated eQTLs colocalize with lead variants from relevant GWAS traits, highlighting a potential path forward for understanding the impact of genetic variation on mechanisms underlying complex traits.

Overall, we demonstrate that while efforts to generate a resource of scRNA-seq data from human tissues[36] are in progress, QTL studies using human bulk RNA-seq data could utilize readily available mouse-derived signature genes to estimate cellular heterogeneity and optimize power to identify cell type-specific genetic associations. As the Tabula Muris resource does not represent all of the human GTEx tissues (28 of 53) it is possible that scRNA-seq resources from other mammalian species could be used to deconvolute the non-represented GTEx tissues. Our study further emphasizes that the straightforward approach of taking tissue heterogeneity into account when conducting genetic association studies has the potential to greatly expand our understanding of the functional impact of genetic variation on molecular and complex human traits.

## Methods

**Single cell expression profiles from 14 mouse organs**. Single cell transcriptome profiles from 14 mouse organs were used in this study[11]. Briefly, transcriptome profiles were generated from three female and four male mice (C57BL/6JN; 10–15-month-old) from: aorta, atrium, bladder, brain nonmicroglia, colon, fat, kidney, liver, mammary gland, muscle, pancreas, skin, spleen, ventricle (Supplementary Table 1). Upon extraction of these organs from the mice, single cell transcriptomes were generated by first sorting by fluorescence-activated cell sorting (FACS) for specific populations (FACS method; SMART-Seq2 RNAseq libraries). We downloaded the normalized gene expression and annotated single-cell clusters from each organ as Seruat[12] R objects (https://figshare.com/articles/Robject_files_for_tissues_processed_by_Seurat/5821263/1).

**Processing scRNA-seq from human liver**. 10X Genomics formatted BAM files from five human total liver homogenate samples[6] were downloaded (GEO accession: GSE11546) and converted to fastq files using 10X bamtofastq (https://support.10xgenomics.com/docs/bamtofastq). Converted fastq files were then processed using cellranger count utility to generate gene expression count matrices, then the five processed liver samples were merged using cellranger aggr utility.

**Annotation of cell populations in human liver scRNA-seq data**. Analysis of scRNA-seq from human liver[6] were conducted following the same approach used to annotate mouse organs[11]. Cells with fewer than 500 detected genes or cells with fewer than 1000 UMI were filtered from the data, resulting in 8119 cells analyzed from human liver. Gene expression was then log normalized and variable genes were identified using a threshold of 0.5 for the standardized log dispersion. PCA was performed on the variable genes and significant PCs. Clustering was performed using a shared-nearest-neighbor graph of the significant PCs and single cells were visualized using UMAP. Cell populations were then annotated based on the expression of known liver marker genes[11].

**Collapsing liver cell population estimates**. To collapse similar cell populations in GTEx liver samples, we examined the UMAP from high-resolution human liver scRNA-seq (Fig. 1b) and compared to the UMAP from low-resolution mouse liver scRNA-seq (Fig. 1c) to identify broader/lower resolution classifications of cell types present in the liver (Supplementary Table 2). We identified populations in the human liver scRNA-seq that were similar (e.g. Hepatocyte populations 0, 1, 3, and 4; Fig. 1b) with a corresponding population in the mouse liver scRNA-seq (e.g. Hepatocyte; Fig. 1c). For populations identified in human not present in mouse, we did not perform any collapsing.

**Annotation of the cell populations in skin scRNA-seq data**. *Human*: Skin scRNA-seq[12] gene expression data and cell annotations for 8388 cells (Supplementary Fig. 3a) were downloaded from http://dom.pitt.edu/rheum/centers-institutes/scleroderma/systemicsclerosiscenter/database/. Cells with fewer than 200 detected genes were filtered from the data. Gene expression was then log normalized and variable genes were identified using a threshold of 0.5 for the standardized log dispersion. PCA was performed on the variable genes and significant PCs. Clustering was performed using a shared-nearest-neighbor graph of the significant PCs and single cells were visualized using UMAP. The single cells were then annotated using provided cell annotations and validated using marker gene expression. As the human skin scRNA-seq contained cell types belonging to various layers of the skin, whereas the mouse scRNA-seq was enriched for epidermal cells, we extracted only the 5670 human cells belonging to the epidermal layer of the skin. We then reanalyzed the subsetted data following the above methods by performing PCA, reclustering, and visualization using UMAP.

*Mouse*: Tabula Muris cell annotations (Supplementary Fig. 3d) were confirmed by examining marker gene expression for epidermal cells, basal cells of the epidermis (Krt1$^{High}$), stem cells of the epidermis (Top2a$^{High}$), leukocytes (Lyz2$^{High}$), and keratinocyte stem cells (Cd34$^{High}$). While Tabula Muris annotated a single keratinocyte stem cell population, we reannotated this population by distinguishing between: (1) inner bulge cell population exhibiting Dkk3$^{High}$ and ITGA6$^{Low}$ expression; and (2) outer bulge cell population exhibiting Fgf18$^{High}$ and ITGA6$^{High}$ expression.

**Deconvolution of bulk tissues using CIBERSORT**. *Identification of signature genes from single cell populations:* For 16 scRNA-seq datasets from human liver, human skin, and scRNA-seq from 14 mouse organs, we obtained gene expression signatures for each annotated cell type (Supplementary Data 1) and used as input into CIBERSORT[3] to estimate the cellular composition of GTEx adult tissues (Supplementary Table 1). For each tissue, we identified differentially expressed genes using Seurat FindMarkers and then extracted the top 200 most significantly overexpressed genes (adjusted $p$-value < 0.05; average log$_2$ fold change >0.25) for each of the annotated scRNA-seq cell types (gene expression signatures). For signature genes obtained from mouse scRNA-seq, we converted the mouse genes to their human orthologs using the biomaRt database[37,38]. The final gene signature sets only included mouse signature genes that also had a human ortholog. For a given signature gene set: (1) if a mouse gene had more than one human ortholog, only one human ortholog was retained in final signature set; and (2) if different mouse genes corresponded to the same human ortholog, only unique human orthologs were retained in the final signature set.

*Cell composition estimation*: The mean expression levels of the signature genes were used as input for CIBERSORT to calculate the relative distribution of the cell populations of 28 GTEx tissues from 14 organs (Supplementary Tables 4 and 5; Supplementary Data 4–17). CIBERSORT (https://cibersort.stanford.edu/) was run with default parameters using the TPM values for the signature genes identified from scRNA-seq in all RNA-seq samples from the analogous GTEx tissue (https://gtexportal.org/home/datasets) (Supplementary Table 1). To determine the cell types detected in GTEx compared to the cell types modeled by mouse, we classified a given cell type as estimable in GTEx as those with CIBERSORT estimates >0.05% in more than 5% of RNA-seq samples from a given GTEx tissue. To estimate cellular heterogeneity across GTEx RNA-seq samples, heterogeneity was measured as the average square distance from the mean for each GTEx tissue. We further examined how time from death or withdrawal of life-support until each tissue sample was fixed/frozen (i.e. ischemic time) is associated with cellular heterogeneity and we did not observe a consistent trend between ischemic time and cellular heterogeneity (Supplementary Fig. 6). GTEx organs are defined as the regions from which tissues are sampled (variable name SMTS from sample attributes data table; phv00169239.v7.p2) and GTEx tissues are defined by the distinct area of the organ where the tissue was taken (variable name SMTSD from sample attributes data table; phv00169241.v7.p2). For example, samples from the GTEx organ, colon, is comprised of two tissues: sigmoid colon and transverse colon.

*Correlation between human and mouse cell type estimates for liver and skin*: Supplementary Data 2 shows, for each combination of cell types from human and mouse, the observed correlation, the mean and standard deviation of correlations in 1000 permutations, the Z-score (calculated as the difference between observed correlation and mean correlation in the permutations, divided by the standard deviation), the empirical $p$-value (calculated using as the number of permutations with correlation greater than the observed value, divided by the number of permutations +1) and the Benjamini–Hochberg-adjusted $p$-value.

**Simulations to test the accuracy of deconvolution**. To test the accuracy of the deconvolution, we conducted simulations to obtain 100 samples of known cell type distributions using scRNA-seq data from human liver. We performed deconvolution on these samples using both human and mouse gene expression signatures and compared the deconvolution results with the known cell type distributions. Specifically, we used the following approach:

1. Using scRNA-seq data from the 8119 human liver cells, we created 100 samples with different cell type compositions. To create each of these pseudo-bulk samples, we selected a random number of cells (between 50 and the total number of cells, using the R function *sample* with *replace = FALSE*) for each cell type. The random number of cells associated with each cell type provides the known sample composition of each pseudo-bulk sample (Supplementary Data 3).
2. To obtain pseudo-bulk expression levels, for each gene we summed the normalized expression (*object@assays$RNA@data* in the Seurat object) of each cell.
3. We ran CIBERSORT using both human and mouse signature genes on all the 100 pseudo-bulk samples. We collapsed cell types as described above (Collapsing liver cell population estimates) and ran CIBERSORT on the collapsed cell types.
4. As a measure of accuracy, we calculated the correlation between the known sample composition, cell types estimated using human signature genes and cell types estimated using mouse signature genes (Supplementary Fig. 2).

**Harmonization of human and mouse scRNA-seq**. To harmonize scRNA-seq from human and mouse liver and skin, mouse genes for each tissue scRNA-seq dataset were first converted to their human orthologs using the BioMart database[37,38]. Mouse and human scRNA-seq were then harmonized by identifying genes that anchor the two datasets using Seurat FindIntegrationAnchors and using these anchors to integrate the datasets using Seurat IntegrateData. Integrated datasets were then visualized using UMAP and corresponding cell types were identified by examining overlap of mouse and human cells.

**eQTL analysis**. To detect eQTLs, we obtained gene TPMs for 153 liver bulk RNA-seq samples and 749 skin bulk RNA-seq samples (sun-exposed and not sun-exposed) from the GTEx V.7 website (https://gtexportal.org/home/) and downloaded WGS VCF files from dbGaP (525 individuals, phs000424.v7.p2). Only genes with TPM > 0.5 in at least 20% samples were considered (19,621 genes in liver and 24,029 in skin). Gene expression data was quantile-normalized independently for each tissue type. For all eQTL analyses, we used the following covariates: age, sex, and the first five genotype principal components (PCs) calculated using 90,081 SNPs in linkage equilibrium[39]. We fitted different linear mixed models (LMMs) using the lme4 package (https://www.jstatsoft.org/article/view/v067i01/0)) to detect eQTLs in liver and skin. We used the following model[18]:

$$\text{Expression} \sim \text{genotype} + \text{covariates} + (1|\text{subject\_id})$$

where (1|subject_id) denotes subject-specific random effects, which we used for skin because several individuals had two samples. For liver, we used sex as random effect to fit an LMM using a method analogous to skin eQTL analysis:

$$\text{Expression} \sim \text{genotype} + \text{covariates} + (1|\text{sex})$$

where (1|sex) denotes sex-specific random effects. We calculated associations with all variants (minor allele frequency >1%) ±1 Mb around each expressed gene. For each gene, we Bonferroni-corrected *p*-values and retained the lead variant. To detect eGenes, we used Benjamini–Hochberg FDR at 10% level on all lead variants.

**Using cell population distributions to improve eQTL detection**. We repeated eQTL detection using LMMs with cellular compositions as covariates. Since several cell types were detected at very low frequency, we only used a subset of the cell types described in Fig. 4. Specifically, we detected liver eQTLs using human (high resolution and collapsed) and mouse (low resolution) cell populations as covariates. We used the following cell populations: (1) for human high resolution: periportal SEC, central venous endothelial cells, gdT cells, hepatocytes0, hepatocytes3, hepatocytes4, inflammatory macrophages, and NK/NKT cells; (2) for human collapsed resolution: endothelial cells, hepatocytes, macrophages, NK cells, B cells, cholangiocytes, and heptatic stellate cells; and (3) for mouse low resolution: endothelial cells of hepatic sinusoid, hepatocytes, Kupffer cells, and NK cells. We detected skin eQTLs using mouse (high and collapsed resolution) cell populations as covariates. We used the following cell populations: (1) for mouse high resolution: epidermis stem cell, leukocyte, inner bulge, outer bulge, epidermis, and epidermis basal cells; and (2) for mouse collapsed resolution: epidermal cells, leukocyte, and inner bulge cells. For each cell population, we compared the following two models:

H₀: expression ~ genotype + covariates + cell_populations + (1|random)
H₁: expression ~ genotype + covariates + cell_populations
　　　+ genotype: cell_population + (1|random)

where (1|random) denotes each tissue's random effect. We next calculated the difference between the two models using ANOVA and obtained $\chi^2$ *p*-values using the pbkrtest package (https://www.jstatsoft.org/article/view/v059i09). For each eGene, we compared each cell population to H₀ and retained only the most significant association. Only eGenes that satisfied two requirements were considered as cell type-associated: (a) Benjamini–Hochberg-adjusted $\chi^2$ *p*-value < 0.1; and (b) $\Delta_{\text{AIC}} = \text{AIC}_{\text{interaction}} - \text{AIC}_{\text{no interaction}} < 0$. eGenes that were associated with only one cell type were considered cell-type-specific. We further determined the impact of cell type abundance on power to detect cell-type-associated eGenes by examining the distribution of $\beta$, standard error, and *p*-value for cell-type-associated-eQTLs between estimated cell types.

**Permutation analysis of liver eQTLs**. To test if the detection of more eQTLs using cell populations as covariates was due to improved accuracy of the linear mixed model estimation or was simply associated with an increased number of covariates, for each top hit (defined as the variant with the strongest *p*-value for each gene), we permuted the cell type distribution across samples, 1000 times. We obtained the average *p*-value, beta and standard error of beta across all permutations and compared these values with the measured *p*-value, beta and standard error of beta for each gene using a paired *t*-test.

**Colocalization of GWAS for skin traits and GTEx skin eQTLs**. For each eGene in the skin eQTL analysis deconvoluted using cell type estimates, we extracted the *p*-values for all variants that were used to perform the eQTL analysis. From the UK BioBank, we obtained summary statistics for 23 skin-related traits (Supplementary Data 25), where the traits were grouped into seven categories based on shared nomenclature in the trait descriptions: (1) malignant neoplasms; (2) melanoma; (3) infection; (4) ulcers; (5) congenital malformations of the skin; (6) other cancer (non-melanoma or malignant neoplasm); and (7) unspecified. For all the variants genotyped in both GTEx and UK BioBank, we used coloc V. 3.1[19] to test for colocalization between eQTLs and GWAS signal. For each colocalization test, we considered only the posterior probability of a model with one common causal variant (PP4). Enrichment of the associations was calculated using a Fisher's Test at multiple PP4 thresholds (0–1; by 0.05 bins), where the contingency table consisted of two classifications: (1) *if* the variant was significantly cell-type-associated (FDR < 0.05); and (2) *if* the variant colocalized with the GWAS trait greater than each PP4 threshold.

**Reporting summary**. Further information on research design is available in the Nature Research Reporting Summary linked to this article.

## Data availability

Sequence data that support the findings of this study (all Figures) is available for human liver scRNA-seq ("GSE11546"); for human skin scRNA-seq (http://dom.pitt.edu/rheum/centers-institutes/scleroderma/systemicsclerosiscenter/database/); and for Tabula Muris mouse scRNA-seq (https://figshare.com/articles/Robject_files_for_tissues_processed_by_Seurat/5821263/1). The source data underlying all Figures is available in Supplementary Tables 1–5 and Supplementary Data 1–25.

## Code availability

Scripts to process, analyze, and generate figures from the data is available at https://github.com/mkrdonovan/gtex_deconvolution.

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

## Acknowledgements

This work was supported in part by a California Institute for Regenerative Medicine (CIRM) grant GC1R-06673 and NIH grants HG008118, HL107442, DK105541, and DK112155. M.K.R.D. was supported by the National Library of Medicine Training Grant T15LM011271.

## Author contributions

K.A.F., M.K.R.D., A.D.C., M.D. conceived the study. M.K.R.D and M.D. performed computational analysis. M.K.R.D. performed scRNA-seq data processing and deconvolution analyses. M.D. performed the eQTL analysis. M.K.R.D. and M.D. performed colocalization analysis. K.A.F. and A.D.C. oversaw the study. M.K.R.D., M.D. and K.A.F. prepared the manuscript.

## Competing interests

The authors declare no competing interests.
