## [Peer Review File · Nature Communications]

Reviewers' Comments:

Reviewer #1:

Remarks to the Author:

Kelly et al., 2019 describe how little work has been published in understanding the cellular composition of rich bioresources such as GTEx bulk RNA-sequencing samples and the utility of explicitly estimating cell type composition and quantifying its impact in a range of analyses, such as inferring eQTLs. To do this, Kelly et al use a novel approach of utilising mouse single cell RNA-sequencing (scRNA-seq) obtained from Tabula Mura to deconvolve human bulk RNA-seq sample mixtures (tissues). They first demonstrate their approach on liver tissue, comparing the use of mouse and human scRNA-seq in accurately inferring cell type composition but also detailing cell types specific to either organisms tissue. This proof-of-concept shows remarkable concordance between human and mouse scRNA-seq estimation of endothelial and hepatocyte populations, whilst poor performance on other cell types (NK-cells) - it's not immediately apparent to me why certain cells would perform so differently across the species. Kelly goes on to estimate cell type composition of all GTEx tissues (that match tabula muris) and demonstrate the power of mapping eQTL as cell * genotype interactions.

Overall the paper is extremely well written and the use of scRNA-seq (particularly from mouse) to perform such a deconvolution is novel. However, I believe the novelty is overemphasised with the omission of relevant citations from previous work performing essentially the same analysis but on a restricted set of GTEx samples and by using purified human cells rather than scRNA-seq.

For example:

(a shameless but very relevant self-plug) - Glastonbury et al., 2019 performed cell type deconvolution using the same method (CIBERSORT) in adipose tissues samples from GTEx and TwinsUK, highlighting that GTEx samples are heterogenous (comparable to this study). Glastonbury et al., 2019 also demonstrated the utility of estimating cell type proportions by quantifying the impact on Transcription wide association analysis (TWAS), cis-eQTL inference and, most importantly, using the estimated cell-types to fit interaction models to discover cell-type specific eQTLs and their overlap with GWAS traits - again, exactly like this study.

Panousis et al., 2019 - Demonstrate the same approach by using CIBERSORT to estimate of cell types in SLE patients, and then by fitting cell * genotype interactions to uncover cell type specific eQTLs.

Additional relevant work are Westra et al., 2015 - a pioneering study that first demonstrated (to my knowledge) the utility of computationally inferring cell type and then fitting interaction models. Kelly et al., would show better scholarship by including these most relevant citations.

Finally, I would also recommend in the spirit of reproducible science, that the code used in this manuscript is made available on Github.

Overall, the paper is novel, interesting and of wide interest to our field and I recommend its publication. I have several additional comments, suggestions and analysis that would strengthen the manuscript which I will detail below.

- 1)Two tissues are particularly well known to contaminate GTEx samples: adipose tissue and blood. For example, most dissected samples such as the aorta or tibial nerve will contain residual adipocytes (as readily observed in GTEx's publicly available histology slide viewer) Could the authors comment on the extent of blood/adipose contamination in GTEx tissues?
- 2)GTEx samples have varying ischemic time (brain vs skin) which affects both sample and RNA integrity. It would be useful to see ischemic vs cell type estimate correlations/confidences.

3) CIBERSORT outputs permutation P-values (FDR) of whether a sample was deconvoluted well - The authors do not seem to use these? Can the authors comment on what fraction of samples are well deconvoluted by CIBERSORT at a suitable FDR (10%)?

4) The interaction models demonstrate the ability to find smaller effect, possibly cell type specific eQTLs. Could the authors please plot or detail for a given tissue, the relationship between a cell's inferred abundance and the cis-eQTL effect size (cell proportion vs interaction coefficient or $-\log_{10}(P\text{-value})$? I.e. Do relatively more abundant cells have larger cis-eQTL effects? This would be to test whether what the authors are seeing is the increased power to detect eQTL for more abundant cells.

5) Through the use of hierarchical clustering or a similar approach is it possible to demonstrate similar tissues have more similar cell type estimates?

6) Please comment more in the discussion of the study's weaknesses. For example, it is possible if there is widespread blood contamination in GTEx tissues, cell type estimates will be inflated as CIBERSORT estimates are relative (normalised to 1 and only consider the cell types modelled) - I.e. you'll get different cell type proportion estimates if you were to drop a cell type or include an additional one for a given tissue. Another point is that you do not know (given the proof-of-concept) what cell types you're inferring incorrectly in all the other non-liver tissues in GTEx.

7) Minor point, the authors should consider color blind friendly color palettes for all their figures.

10.1136/annrheumdis-2018-214379

10.1016/j.ajhg.2019.03.025

10.1371/journal.pgen.1005223

- Craig A. Glastonbury (BenevolentAI)

Reviewer #2:

Remarks to the Author:

The paper describes a conceptual pipeline for discovering human cell-type specific eQTLs, based on cell-type quantities, predicted using mouse single cell RNA-seq. The authors used the recently published Tabula Muris database as their single cell atlas for investigating the cell-type repertoire in the GTEx database, which contains different tissues, collected from hundreds of human individuals. However, the paper does not provide evidence for the accuracy of the computational pipeline and predictions. The results are therefore shown from different aspects of cell deconvolution and eQTL calculation without providing a conceptual proof for the accuracy of prediction.

Major comments:

1. The manuscript does not show that the cell-type quantity prediction was accurate. I could not conclude that the cell quantities were similar between human and mouse based on the results shown (see below). If data is simulated based on the human single cell RNA-seq, it can be shown that it can be predicted using the mouse data. For example, setting the level of each cell type, adding noise to test the quality of the pipeline, and then running and comparing the deconvolution with either the mouse single cell, the human single cell, and a random single cell data (by shuffling the labels of the cell types). Such analysis may enable to test the actual effect of deconvolving with mouse data.

2. In figure 2D, it seems like there are high positive and negative correlations between the quantities of many cell types, not necessarily the obvious ones. For example, there is a positive correlation of mouse endothelial cells with human hepatocytes type 4 and all the mouse cell types are positively correlated with most human cells. Why is that? It seems like all the other cell types, except for the hepatocytes, are very similar in their quantities. It might be an artifact due to the large effect of the hepatocytes. A comparison between the cell type quantities patterns within each organism may explain the observed values. Another option is to use human published datasets, containing both the bulk RNA-seq and cell type compositions; these datasets can be used to test

the deconvolution accuracies using single cell data from mice compared to the appropriate human data (if available).

3. Its not fully explained how the enrichment scores were calculated for the cell type associations. What was the background in this analysis? Did it take into account variants and traits from the same group that were not co-localized?

4. It is not obvious why the AIC criterion was used in the first place for the selection of cell population-specific, as the number of variables is equal between different eGenes. Specifically, why $AIC < 0$ was used? Was this selected arbitrary? Can it be shown that the results are robust to the selection of parameters.

5. IN general, I do not see what is the "bottom line" novel biological insights from this work. Can you explain this better in the text, or make additional "meta-analysis" to derive general biological principles?

Minor comments

1. Whereas the title and abstract of the paper emphasizes systematic analysis of cell-type-associated variants, it seems that the actual focus of the manuscript is on the integration of mouse data for deconvolution of human data.

2. Previous methods in the field, using deconvolved cell type quantities for eQTL discovery, such as Westra et al. 2015, was not discussed (and cited).

3. In reference 11, the human and mouse liver single cell RNA-seq were aligned before using them both. I would expect that such alignment would be also needed before applying deconvolution in this work. Different alignment/harmonization methods are available, how do the results change with different cell alignment methods?

4. In figure 3B, the meaning of "cell type was not detected" is not clear. Are not all cell types being predicted by the deconvolution?

5. Line 264: "Furthermore, using cell populations as covariates resulted in decreased effect size (β) (Figure 4C) and standard error (SE) of β (Figure 4D)". Isn't it trivia? - when using more covariates, the coefficient of the factor decreases and the error is lower (because we have more factors to explain the SNP). It is not clear that the information of cell population provides the observed advantage. To show the advantage in using the cell population values as covariates, it is possible to show superiority compared to the advantage obtained when using permuted cell population data as covariates.

6. In line 295, did you mean "Skin" instead of "Liver"?

We would like to thank the Reviewers for the contribution of their extremely valuable comments and suggestions. We have incorporated the comments (italicized below) into the manuscript and feel that it has been greatly improved. Overall, the Editor and Reviewers primarily requested to see more extensive validation of the GTEx tissues cellular composition predictions. To do this, we included skin as a second proof-of-concept tissue to determine how estimates of cellular composition of GTEx skin compared between those deconvoluted using mouse skin scRNA-seq and human skin scRNA-seq. We found, similar to liver, deconvolution performed using mouse scRNA-seq produced similar results to using human scRNA-seq, thus supporting mouse scRNA-seq as a suitable alternative for the deconvolution of heterogeneous tissues. Please find the details of this and our other modifications below and highlighted in an accompanying document.

Reviewer's comments:

Reviewer #1: *Kelly et al., 2019 describe how little work has been published in understanding the cellular composition of rich bioresources such as GTEx bulk RNA-sequencing samples and the utility of explicitly estimating cell type composition and quantifying its impact in a range of analyses, such as inferring eQTLs. To do this, Kelly et al use a novel approach of utilizing mouse single cell RNA-sequencing (scRNA-seq) obtained from Tabula Mura to deconvolve human bulk RNA-seq sample mixtures (tissues).*

- 1. They first demonstrate their approach on liver tissue, comparing the use of mouse and human scRNA-seq in accurately inferring cell type composition but also detailing cell types specific to either organisms' tissue. This proof-of-concept shows remarkable concordance between human and mouse scRNA-seq estimation of endothelial and hepatocyte populations, whilst poor performance on other cell types (NK-cells) - it's not immediately apparent to me why certain cells would perform so differently across the species. Kelly goes on to estimate cell type composition of all GTEx tissues (that match tabula muris) and demonstrate the power of mapping eQTL as cell * genotype interactions.*

We thank the Reviewer for this comment and agree that discussing the differences in estimation performance across cell types between species is an important point to make. Discrepancies in cell composition estimates between the species may be occurring for a multitude of reasons, primarily resulting from 1) technical and 2) biological differences, described in detail below.

- 1. Technical differences** influence our ability to distinguish between similar cell types (i.e. cell type resolution), and thus impact the resolution at which cellular composition can be estimated. These technical differences include: 1) the number of cells captured and subjected to scRNA-seq, which may influence the proportion of observed common or rare cell types; and 2) how the tissue was sampled, which may capture how populations are distinguished by spatial location (i.e. zonation). For example, we note that high resolution human scRNA-seq (total liver homogenate; 8,119 cells; 15 cell types) captured many more, but similar cell types as low resolution mouse scRNA-seq (purification of hepatocyte and non-parenchymal cells prior to FACS sorting based on viability; 710 cells; 5 cell types), highlighting differences in the number of cells and tissue sampling. We identified four zone-specific hepatocyte populations and two zone-specific endothelial cell populations in human liver scRNA-seq, whereas in mouse liver scRNA-seq, we identified a single hepatocyte and endothelial population. We found that the mouse hepatocyte population estimates were highly correlated with one hepatocyte zone and the mouse endothelial cell population estimates were correlated with both human endothelial populations. Similarly, in human epidermis scRNA-seq (digested dorsal forearm skin biopsies; 5,670 cells; 9 cell types), we identified several keratinocyte subpopulations, whose estimates corresponded to estimates from distinct epidermal cell populations identified in mouse skin (FACS sorted epidermal keratinocytes; 2,263 cells; 6 cell types), however other human epidermal cell types (e.g. fibroblasts) were not correlated with any mouse population estimates, as the mouse populations were sorted for keratinocytes. This would suggest cell type concordance is largely influenced by technical differences

that exist in scRNA-seq methodology (e.g. number of cells analyzed and how the tissue was sampled), rather than species differences.

2. Biological differences, such as known cellular differences between species, impact estimation performance. We observed estimates from immune cells were not well correlated between mouse and human in both liver and skin estimates. It has been shown that humans and mice have immunological differences and that a given immune cell response/function in humans may not occur in the same way in mice.

In summary, differences in cell estimates between human and mice can be attributed to several factors, mainly pointing to technical differences in experimental approach and subtle immune differences known to exist between humans and mice. To explain the causes of these differences between human and mouse estimates better in the manuscript, we have added additional text to several sections in the Results: 1) scRNA-seq from murine and human analogous tissues capture similar cell types; 2) Mouse liver cell populations can estimate cellular composition of human liver GTEx samples; and the new section 3) Deconvolution of GTEx skin using mouse skin scRNA-seq confirms mouse can signature genes can estimate cellular composition). We have also added a new section in the Discussion (pg. 27).

2. *Overall the paper is extremely well written and the use of scRNA-seq (particularly from mouse) to perform such a deconvolution is novel. However, I believe the novelty is overemphasized with the omission of relevant citations from previous work performing essentially the same analysis but on a restricted set of GTEx samples and by using purified human cells rather than scRNA-seq. For example: (a shameless but very relevant self-plug) - Glastonbury et al., 2019 performed cell type deconvolution using the same method (CIBERSORT) in adipose tissues samples from GTEx and TwinsUK, highlighting that GTEx samples are heterogenous (comparable to this study). Glastonbury et al., 2019 also demonstrated the utility of estimating cell type proportions by quantifying the impact on Transcription wide association analysis (TWAS), cis-eQTL inference and, most importantly, using the estimated cell-types to fit interaction models to discover cell-type specific eQTLs and their overlap with GWAS traits - again, exactly like this study. Panousis et al., 2019 - Demonstrate the same approach by using CIBERSORT to estimate of cell types in SLE patients, and then by fitting cell * genotype interactions to uncover cell type specific eQTLs. Additional relevant work is Westra et al., 2015 - a pioneering study that first demonstrated (to my knowledge) the utility of computationally inferring cell type and then fitting interaction models. Kelly et al., would show better scholarship by including these most relevant citations.*

These are indeed important citations to include in this work and we thank the Reviewer for pointing out our oversight. The work by Glastonbury *et al.* highlights the necessity to account for the cellular complexity of bulk tissues in genetic association studies and in understanding disease and has been cited in the Introduction (Pg. 3). The work by Panousis *et al* showed the use of cell type abundances as interaction terms and has been cited in the Results (Pg. 21). Westra *et al.* is a substantial work that uses cellular composition to identify cell-type-associated eQTLs and has been cited in the Results (Pg. 21).

3. *Finally, I would also recommend in the spirit of reproducible science, that the code used in this manuscript is made available on Github.*

We agree with open-access to code used to analyze data and generate figures. You can find the repository of code for this manuscript at: www.github.com/mkrdonovan/gtex_deconvolution. Additionally, we have included a Data Availability section (pg. 36) to describe from where we downloaded sequencing data and the link to GitHub for scripts used for data processing, analysis, and figure generation.

4. *Overall, the paper is novel, interesting and of wide interest to our field and I recommend its publication. I have several additional comments, suggestions and analysis that would strengthen the manuscript which I*

will detail below. Two tissues are particularly well known to contaminate GTEx samples: adipose tissue and blood. For example, most dissected samples such as the aorta or tibial nerve will contain residual adipocytes (as readily observed in GTEx's publicly available histology slide viewer) Could the authors comment on the extent of blood/adipose contamination in GTEx tissues?

While the Reviewer proposes an interesting question by asking the extent to which blood and adipose contaminate GTEx tissues, we acknowledge that this is a difficult question to answer. Cellular deconvolution of GTEx bulk tissues using CIBERSORT are dependent on the gene signatures of cell types known to reside in that tissue, as measured by scRNA-seq. We are thus limited to only estimating the contribution of cell types identified by scRNA-seq, which we show is largely influenced by technical approach (please see: Reviewer #1, comment #1). While it might seem intuitive that gene signatures for adipose cell types or various blood cell types, theoretically obtained from scRNA-seq from adipose or whole blood, could be used in combination with signature genes identified from scRNA-seq for a given tissue to perform cellular deconvolution, we emphasize that similar cell types across different tissues serve distinct functions and thus have distinct gene expression profiles. For example, adipose residing close to different organs adopt distinct functions, such as adipocytes stimulating organ remodeling and regeneration in skin, bone marrow, and mammary glands or the communication between hair follicles and skin adipocytes in promoting epithelial stem cell quiescence/activation. It is thus unclear what is a tissue contaminant and what is an unrecognized resident cell type or even zone-specific to that tissue not measured by scRNA-seq, and we thus are hesitant to estimate their contribution to cellular heterogeneity without gene signatures obtained from scRNA-seq in the corresponding tissue. The question of how comparable gene expression signatures are from similar cell types across tissues, however, is an intriguing question, albeit out of the scope of this work, but would be of interest to pursue in the future.

- 5. GTEx samples have varying ischemic time (brain vs skin) which affects both sample and RNA integrity. It would be useful to see ischemic vs cell type estimate correlations/confidences.*

We agree that a useful analysis is examining how cellular heterogeneity for a given sample is associated with the time from death or withdrawal of life-support until the tissue sample is fixed/frozen (i.e. ischemic time), because ischemic time can be quite variable across samples from a tissue and there may be changes in gene expression occurring as a result of death. For each GTEx RNA-seq sample, we calculated the correlation in the sample's heterogeneity, as measured by the correlation of the average square distance from the mean, with the sample's ischemic time (Supplemental Figure 5). We observed that in some tissues, ischemic time is modestly positively correlated with heterogeneity (atrium, brain, fat, mammary, and ventricle), in other tissues, ischemic time is negatively correlated with heterogeneity (liver, colon, and muscle), and in some tissues, there is no correlation between ischemic time and heterogeneity (aorta, bladder, spleen, kidney, pancreas, and skin). While it may be possible that increased ischemic time impacts gene expression, whether by compromising the integrity of the tissue/RNA or by altering normal gene expression levels expected in living individuals, we did not observe a consistent trend between ischemic time and cellular heterogeneity. To address how cellular heterogeneity is related to ischemic time, we have included additional text in our Methods (pg. 32) and as Supplemental Figure 5.

- 6. CIBERSORT outputs permutation P-values (FDR) of whether a sample was deconvoluted well - The authors do not seem to use these? Can the authors comment on what fraction of samples are well deconvoluted by CIBERSORT at a suitable FDR (10%)?*

We thank the Reviewer for the suggestion to discuss the p-values for the CIBERSORT permutations. For each of the 14 tissues, we found that all samples were well-deconvoluted (P-value < 0.001). To clarify this point, we have added a sentence to the Results (pg. 16) and updated the Supplemental Tables 5–20 to include the CIBERSORT P-values.

7. *The interaction models demonstrate the ability to find smaller effect, possibly cell type specific eQTLs. Could the authors please plot or detail for a given tissue, the relationship between a cell's inferred abundance the cis-eQTL effect size (cell proportion vs interaction coefficient or $-\log_{10}(P\text{-value})$)? I.e. Do relatively more abundant cells have larger cis-eQTL effects? This would be to test whether what the authors are seeing is the increased power to detect eQTL for more abundant cells.*

We thank the Reviewer for this insightful question and agree that it is important to ask if cell abundance contributes to power to detect cell-type-associated eQTLs. To answer this, we investigated if cell abundance is associated with effect size and found that cell abundance did not influence power to detect cell-type-associated eQTLs. Specifically, to understand if cell abundance contributed to power, we asked how the distribution of β , standard error (SE), and P-value for cell-type-associated-eQTLs changes between cell types (Supplemental Figure 8A-C). We found, despite substantial differences in liver cell abundances, absolute effect size, SE, and significance of eQTL detection did not change between cell types. We performed a similar analysis for each cell type estimated in skin and we also observed that cell type abundance did not appear to influence our power to detect cell-type-associated eQTLs (Supplemental Figure 8D-F). We have added these findings to the Results (pg. 21), Methods (pg. 34), and Supplemental Figure 8.

8. *Through the use of hierarchical clustering or a similar approach is it possible to demonstrate similar tissues have more similar cell type estimates?*

We agree this suggested analysis is an important sanity check to confirm that similar tissues cluster together, as would be expected if tissues share similar cell type compositions. To address this, we generated an UMAP using the expression of the signature genes from 14 mouse tissue types across all RNA-seqs from the 28 GTEx tissues and asked how well these genes distinguished between the GTEx tissues (Figure 3A). We found, as the Reviewer predicted, that similar tissues clustered closely together, indicating similar tissues likely share cell types. Specifically, this sanity check confirmed that we observed expected tissues clustering together, including the heart tissues (atrium and ventricle), brain tissues (cortex, frontal cortex, hippocampus, anterior cingulate cortex, amygdala, substantia nigra, spinal cord, putamen, nucleus accumbens, caudate, and hypothalamus), adipose (visceral and subcutaneous), and colon (sigmoid and transverse). Of note, within the brain we also observe clustering according to zonation, including clustering of samples from the cerebellum and cerebellar hemisphere, as well as clustering of samples from the frontal cortex, cortex, and anterior cingulate cortex. Together this analysis demonstrates that the expression of the signature genes used for deconvolution distinguish between the different GTEx tissues, as well as show the expression of the signature genes are similar between similar tissues, supporting the observation that similar tissues have similar cell type compositions. We have included this analysis in the Results (pg. 15) and as Figure 4A.

9. *Please comment more in the discussion of the studies weaknesses. For example, it is possible if there is widespread blood contamination in GTEx tissues, cell type estimates will be inflated as CIBERSORT estimates are relative (normalized to 1 and only consider the cell types modelled) - I.e. you'll get different cell type proportion estimates if you were to drop a cell type or include an additional one for a given tissue. Another point is that you do not know (given the proof-of-concept) what cell types you're inferring incorrectly in all the other non-liver tissues in GTEx.*

We thank the Reviewer for this comment and agree that the cell composition estimates we obtain from CIBERSORT are dependent on the signature genes from cell types that were assayed by scRNA-seq, which speaks to the Reviewer's first comment (see Review #1, comment #1), where we highlight that estimation performance is influenced by technical and biological factors. Additionally, as the Reviewer noted, this estimation approach is limited to only estimating cell types identified by scRNA-seq, where it may be possible that we are not estimating potential cell contaminants or other uncharacterized cell types (please see Reviewer #1, response #4) and thus cell estimates may be inflated (or reduced) relative to the true

cellular composition. We believe these weaknesses to this study stresses the need for a completed human scRNA-seq resource, such as the Human Single Cell Atlas, that maps the cellular composition across human tissues. Completion of this resource will help us better define the signatures genes of common, rare, zonation-specific, and yet-to-be discovered cell types, for which cell composition estimates by deconvolution of RNA-seq from bulk tissues will be improved. To further address weaknesses of this study, we have added additional text Discussion (pg. 27).

10. Minor point, the authors should consider color blind friendly color palettes for all their figures.

We appreciate this comment and have included an accompanying colorblind-friendly figure to Figure 4B in Supplemental Figure 4.

Reviewer #2: *The paper describes a conceptual pipeline for discovering human cell-type specific eQTLs, based on cell-type quantities, predicted using mice single cell RNA-seq. The authors used the recently published Tabula Muris database as their single cell atlas for investigating the cell-type repertoire in the GTEx database, which contains different tissues, collected from hundreds of human individuals. However, the paper does not provide evidence for the accuracy of the computational pipeline and predictions. The results are therefore shown from different aspects of cell deconvolution and eQTL calculation without providing a conceptual proof for the accuracy of prediction.*

1. The manuscript does not show that the cell-type quantity prediction was accurate. I could not conclude that the cell quantities were similar between human and mouse based on the results shown (see below). If data is simulated based on the human single cell RNA-seq, it can be shown that it can be predicted using the mouse data. For example, setting the level of each cell type, adding noise to test the quality of the pipeline, and then running and comparing the deconvolution with either the mouse single cell, the human single cell, and a random single cell data (by shuffling the labels of the cell types). Such analysis may enable to test the actual effect of deconvolving with mouse data.

We appreciate this comment and agree that additional analyses demonstrating the accuracy of cell-type quantity prediction would be helpful to better support our conclusions. To address the Reviewer's concern, we have expanded the presented analyses to clarify the results, which better depicts the similarity in mouse and human cell estimates, as responded to in a similar comment (please see Reviewer #2, response #2). Further, while the Reviewer proposed an elegant experiment using simulation to test the performance of deconvolution using human versus mouse scRNA-seq, by request of the Editor, we thought this concern could be alternatively addressed by including a second proof-of-concept tissue.

To demonstrate the accuracy of using mouse scRNA-seq for deconvolution, in addition to liver, we used skin as a second proof-of-concept tissue. Using signature genes from cell types obtained from human epidermal scRNA-seq (5,670 cells; nine cell types) (Figure 3A,B), we estimated the cellular composition of GTEx skin (Figure 3G). While we had previously used mouse skin scRNA-seq for deconvolution, we are particularly thankful for this suggestion, as it allowed us to identify two distinct mouse cell populations (Inner and Outer bulge) that had been previously defined as a single population (keratinocyte stem cells) (Supplemental Figure 2D,E). Using the reannotated mouse scRNA-seq (2,263 cells; six cell types) (Figure 3C,D) to re-estimate the cellular composition of GTEx skin (Figure 3H), we observed that the human and mouse skin cell estimates were highly correlated (Figure 3I-K; Supplemental Figure 3). Similar to liver, the main discrepancies were the result of technical differences (number of cells analyzed and tissue sampling methodology) impacting cell type resolution and immune cell estimate differences (please see Reviewer #1, comment #1). Additionally, as a consequence of having re-estimated the cellular composition of GTEx skin using six cell types from mouse scRNA-seq, we reanalyzed the eQTLs from GTEx skin and re-performed the colocalization analysis.

These new analyses resulted in a new Results section (Deconvolution of GTEx skin using mouse skin scRNA-seq confirms mouse signatures gene can estimate cellular composition), in a new Figure 3, in Supplemental Figures 2 and 3, in modifications to the text (Sections: 1) eQTL analysis of deconvoluted GTEx skin confirms ability to identify cell-type-specific regulatory variants, and 2) Colocalization identifies cell-type-specific regulatory variants are associated with specific skin diseases), in updates to Figures 4, 5, and 6, and in updates to the Discussion (pg. 27).

2. In figure 2D, it seems like there are high positive and negative correlations between the quantities of many cell types, not necessarily the obvious ones. For example, there is a positive correlation of mouse endothelial cells with human hepatocytes type 4 and all the mouse cell types are positively correlated with most human cells. Why is that? It seems like all the other cell types, except for the hepatocytes, are very similar in their quantities. It might be an artifact due to the large effect of the hepatocytes. A comparison

between the cell type quantities patterns within each organism may explain the observed values. Another option is to use human published datasets, containing both the bulk RNA-seq and cell type compositions; these datasets can be used to test the deconvolution accuracies using single cell data from mice compared to the appropriate human data (if available).

We thank the Reviewer for pointing out that Figure 2D (now Figure 2F) is unclear. The intention behind using a heatmap depicting the correlation between all pairwise human and mouse cell composition estimates was to condense the information and highlight high correlation between hepatocyte estimates (Figure 2G) and endothelial cells (Figure 2H). We agree that it is important to address the other unexpected correlations we observe in manuscript Figure 2D (now Figure 2F). To better describe this data we have included the raw scatter plots comparing the cell type quantities between human and mouse, similar to the plots shown in Figure 2G and Figure 2H as supplemental figures (liver, Supplemental Figure 1; skin, Supplemental Figure 3), which shows, despite apparent unexpected correlation between mouse and human estimates, like human Hepatocyte 4 and mouse Endothelial cells, these comparisons are not nearly as striking as matched cell types. Overall, we observe the highest (and most significant) correlations between analogous human and mouse cell types, which is corroborated by a new harmonization of mouse and human scRNA-seq analysis (please see Reviewer #2, response #8).

3. *It's not fully explained how the enrichment scores were calculated for the cell type associations. What was the background in this analysis? Did it take into account variants and traits from the same group that were not co-localized?*

We thank the Reviewer for these questions and have clarified how enrichment scores were calculated in the methods. To calculate enrichment of each GWAS trait for cell-type-associated eQTLs we used a Fisher's Test, where the contingency table consists of two classifications: 1) *if* the variant was significantly cell-population specific (FDR < 0.05), and 2) *if* the variant colocalized with the GWAS trait greater than each posterior probability 4 (PP4) threshold. We thus calculated enrichment for each GWAS trait independently and took all variants into account. Clarification of the enrichment calculation methods can be found in the Methods pg. 35.

4. *It is not obvious why the AIC criterion was used in the first place for the selection of cell population-specific, as the number of variables is equal between different eGenes. Specifically, why $AIC < 0$ was used? Was this selected arbitrary? Can it be shown that the results are robust to the selection of parameters.*

We are very grateful for this comment, as it points out that we made an error in the text describing AIC criterion, where we should have written $\Delta_{AIC} < 0$, and not $AIC < 0$. In addition to fixing this error identified by the Reviewer in our writing, we additionally agree that the rationale for the criterion used for selection of cell population-specific eQTLs is not explained with enough detail. We performed the following for each gene and each cell type:

- 1) We used two models (with and without the interaction term "gene:cell type")
- 2) For both models, we calculate the Akaike Information Criterion (AIC), as a measure of the quality of each model.
- 3) Given that: a) we applied multiple models to the same data; and b) AIC is a measure of how well the model fits the data, we determine that an eGene is cell population-associated if the AIC of the model with the interaction term "gene:cell type" is significantly lower than the AIC for the model without the interaction term.
- 4) We calculated whether the two models were different using ANOVA and obtained χ^2 p-values using the pbkrtest package (<https://www.jstatsoft.org/article/view/v059i09>)
- 5) Only eGenes that satisfied two requirements were considered as cell population-associated: a) Benjamini-Hochberg-adjusted χ^2 p-value < 0.1; and b) $\Delta_{AIC} = AIC_{interaction} - AIC_{no\ interaction} < 0$

- Akaike, H. (1973), "Information theory and an extension of the maximum likelihood principle", in Petrov, B. N.; Csáki, F. (eds.), 2nd International Symposium on Information Theory, Tsahkadsor, Armenia, USSR, September 2-8, 1971, Budapest: Akadémiai Kiadó, pp. 267–281. Republished in Kotz, S.; Johnson, N. L., eds. (1992), Breakthroughs in Statistics, I, Springer-Verlag, pp. 610–624.

We have changed a sentence of the Methods (section “Using cell population distributions to improve eQTL detection”; pg. 34) to Only eGenes that satisfied two requirements were considered as cell population-associated: a) Benjamini-Hochberg-adjusted χ^2 p-value <0.1 ; and b)

$$\Delta_{AIC} = AIC_{interaction} - AIC_{no\ interaction} < 0 .$$

5. *IN general, I do not see what is the "bottom line" novel biological insights from this work. Can you explain this better in the text, or make additional "meta-analysis" to derive general biological principles?*

We would like to clarify that this work had two motivating goals: 1) to understand if mouse scRNA-seq could be used as an alternative to human scRNA-seq for the cellular deconvolution of GTEx RNA-seq data, and 2) to determine if using cell composition estimates as covariates in eQTL analyses can identify novel associations and the nature of the associations. Specifically, it is known that functional genetic variation alters gene expression distinctly across cell types, however most eQTL studies focus on examining the impact of genetic variation on gene expression in tissues, as reported by GTEx. To perform cell-type-specific eQTL analyses, it is possible to deconvolute the cellular composition of bulk RNA-seq using signature genes from scRNA-seq generated from analogous tissues. While we and others have shown this approach is possible, human scRNA-seq is not yet available for all tissues of interest, thus we proposed mouse scRNA-seq as a potential alternative data source. Our study demonstrates that readily available mouse gene expression signatures can be used to deconvolute the cellular composition of human tissues and the estimation of cellular heterogeneity by deconvolution enhances the genetic insights yielded from the GTEx resource. To clarify the motivation of this work, we have added additional text to the Abstract (pg. 2) and Introduction (pg. 4).

6. *Whereas the title and abstract of the paper emphasizes systematic analysis of cell-type-associated variants, it seems that the actual focus of the manuscript is on the integration of mouse data for deconvolution of human data.*

We realized based on the above comment (please see Reviewer #2, comment #5) that we could improve the description of the motivations of this work to emphasize both the use of mouse scRNA-seq for cellular deconvolution of GTEx tissues and the use of these estimates of cellular heterogeneity for characterizing the functional impact of cell-type-specific genetic variation. To clarify the motivation of this work, we have added additional text to the Abstract (pg. 2) and Introduction (pg. 4).

7. *Previous methods in the field, using deconvolved cell type quantities for eQTL discovery, such as Westra et al. 2015, was not discussed (and cited).*

The absence of this important reference was also noted by Reviewer #1 (please see Reviewer #1, comment #2) and has been discussed and cited in the Results (Pg. 21).

8. *In reference 11, the human and mouse liver single cell RNA-seq were aligned before using them both. I would expect that such alignment would be also needed before applying deconvolution in this work. Different alignment/harmonization methods are available, how do the results change with different cell alignment methods?*

We agree with the reviewer that it is important to understand how the cell populations from mouse and human align with each other after harmonization. We had not previously attempted harmonization and are thankful for this suggestion, in performing this analysis we realized it served as an important validation step in confirming similarity of mouse and human cell populations. To analyze how populations from human and

mouse scRNA-seq align, we performed integration and observed the cell types with correlated estimates also clustered together. The analytical strategy used for integration, called canonical correlation analysis (CCA) presented by Butler *et al.* (reference 11), is able to harmonize datasets from different sources, such as cross-platform and cross-species. Given the unique capability of CCA, we were able to integrate scRNA-seq data generated human and mouse liver and skin scRNA-seq to identify shared populations between the datasets (Figure 2A,B; Figure 3E,F). To address the results of these harmonization analyses of mouse and human scRNA-seq, we have added additional text to the Results (pg. 7 and 11), Methods (new section: “Harmonization of human and mouse scRNA-seq”), and as new panels in Figure 2A,B and Figure 3E,F.

9. *In figure 3B, the meaning of “cell type was not detected” is not clear. Are not all cell types being predicted by the deconvolution?*

We thank the Reviewer for this comment and agree that this statement needs to be clarified in the text. In Figure 3B (now Figure 4C), we show a bar plot comparing the number of cell types identified in mouse scRNA-seq (light grey) and the number of these cell types that were estimable for each GTEx tissue, where technical or biological factors may be influencing our ability to use mouse gene signatures to estimate cellular composition, as discussed in comment #1 from Reviewer #1. To measure the number of cell types identified “in mouse scRNA-seq”, we counted the number of cell types identified for each tissue by Tabula Muris scRNA-seq. To measure the number of cell types that were estimable for each human GTEx tissue, we examined the CIBERSORT results for each cell type from mouse and a cell type was considered “detected in GTEx” if the cell type comprised greater than 0.05% in more than 5% of RNA-seq samples from a given GTEx tissue. This cutoff was selected to identify the cases where the cell type was not detected in most samples. Clarification of “GTEx cell type detection” can be found in the Results (pg. 16) and Methods (pg. 32).

10. *Line 264: “Furthermore, using cell populations as covariates resulted in decreased effect size (β) (Figure 4C) and standard error (SE) of β (Figure 4D)”. Isn’t it trivia? - when using more covariates, the coefficient of the factor decreases and the error is lower (because we have more factors to explain the SNP). It is not clear that the information of cell population provides the observed advantage. To show the advantage in using the cell population values as covariates, it is possible to show superiority compared to the advantage obtained when using permuted cell population data as covariates.*

We thank the Reviewer for this important suggestion to confirm inclusion of cell population values as covariates increased the number of identified eQTLs due to improved accuracy of the model, rather than simply associated with an increased number of covariates. To address this, for each top hit (defined as the variant with the strongest p-value for each gene), we permuted the cell population distribution across samples 1,000 times. We obtained the average p-value, beta and standard error of beta across all permutations and compared these values with the measured p-value, beta, and standard error of beta for each gene using t-test (Supplemental Figure 7) and found that the increase in the number of eQTLs detected was not due to using more covariates. This analysis has been incorporated into text in the Results (pg. 21), in the Methods (section “Permutation analysis of liver eQTLs”; pg. 34), and as Supplemental Figure 7.

11. *In line 295, did you mean “Skin” instead of “Liver”?*

We thank the Reviewer for catching this error. We have updated the text to reflect this correction.

Reviewers' Comments:

Reviewer #1:

Remarks to the Author:

I believe the authors have answered all my questions and I think this manuscript should be published.

Reviewer #2:

Remarks to the Author:

The paper was improved by showing some links between single cells from human and mouse origin in both Liver and Skin tissue. I believe that the current version is more suitable for publication than the original one.

However, there is still a major issue, raised in my original comments (original comment #1) that was not addressed by the authors. The main aim of this paper is to apply mouse-based deconvolution on human bulk samples, using tissues for which single cell RNA-seq data is not available. There are noticeable differences between cell types in the two species. This was explained as biological and technical variations between the datasets. However, the CCA analysis shows quite strikingly that the mouse single cell data does not cover most of the cell types in the human data. While the collapsed regions seem to be shared by the two species, in most cases there is an obvious separation between human and mouse regions. Given the dissimilarities between the species, there is no reason to believe that the cell composition, inferred by the deconvolution process, is justified.

Additional comments, related to the revised manuscript:

1. The p-values used in Figures 2F and 3I are misleading. It looks like everything is extremely significant although the correlation coefficients are very low in most cases. This is probably due to large sample sizes. A solution might be to permute the cell compositions of each of the cell types many times, calculating the correlations of the permuted compositions with the original compositions. Then, to calculate the permutation p-value for each cell type by comparing its original correlation level to its permutations.
2. Figure 4 is named figure 3 in the manuscript file.
3. Line 366: "By performing a permutation test, we excluded that the detection of a larger number of eQTLs using cell populations was due to the fact that a larger number of covariates was used". That's confusing - it can be interpreted as an increase in the number of eQTLs just due to the addition of power by adding covariates and not necessarily biological meaningful covariates.

Reviewer #2 (Remarks to the Author):

The paper was improved by showing some links between single cells from human and mouse origin in both Liver and Skin tissue. I believe that the current version is more suitable for publication than the original one. However, there is still a major issue, raised in my original comments (original comment #1) that was not addressed by the authors. The main aim of this paper is to apply mouse-based deconvolution on human bulk samples, using tissues for which single cell RNA-seq data is not available. There are noticeable differences between cell types in the two species. This was explained as biological and technical variations between the datasets. However, the CCA analysis shows quite strikingly that the mouse single cell data does not cover most of the cell types in the human data. While the collapsed regions seem to be shared by the two species, in most cases there is an obvious separation between human and mouse regions. Given the dissimilarities between the species, there is no reason to believe that the cell composition, inferred by the deconvolution process, is justified.

Original comment:

The manuscript does not show that the cell-type quantity prediction was accurate. I could not conclude that the cell quantities were similar between human and mouse based on the results shown (see below). If data is simulated based on the human single cell RNA-seq, it can be shown that it can be predicted using the mouse data. For example, setting the level of each cell type, adding noise to test the quality of the pipeline, and then running and comparing the deconvolution with either the mouse single cell, the human single cell, and a random single cell data (by shuffling the labels of the cell types). Such analysis may enable to test the actual effect of deconvolving with mouse data.

We apologize for misinterpreting the original comment by the Reviewer and thank him/her for clarifying. We believe the following analysis greatly strengthened our manuscript and we appreciate the Reviewer's insightful input. To test the accuracy of deconvolution, we conducted simulations to obtain samples of known cell type distributions using scRNA-seq data from human liver. We used the following approach (described in the Methods section "Simulations to compare deconvolution using mouse and human signature genes" and in Figure S2) to perform simulations and test the accuracy of deconvolution using human and mouse gene expression signatures.

1. Using scRNA-seq data from the 8,119 human liver cells, we created 100 samples with different cell type compositions. To create each of these pseudo-bulk samples, we selected a random number of cells (between 50 and the total number of cells, using the R function *sample* with *replace = FALSE*) for each cell type. The random number of cells associated with each cell type provides the known sample composition of each pseudo-bulk sample (Table S5).
2. To obtain pseudo-bulk expression levels, for each gene we summed the normalized expression (`@assays$RNA@data` in the Seurat object) of each cell.
3. We ran CIBERSORT using both human and mouse signature genes on all the 100 pseudo-bulk samples. We also obtained collapsed cell types by combining cell types as described in the Methods section "Collapsing liver cell population estimates".
4. We calculated the correlation between the known sample composition, cell types estimated from human signature genes, and cell types estimated from mouse signature (Figure S2).

We observed that the correlation between the known cell composition and estimates using human signature genes is very high (>0.75) for all cell types except for stellate cells ($r = 0.573$). Of note, stellate cell signature

genes were based on only 79 cells. From mouse scRNA-seq, we obtained five cell types and, for four of them (endothelial, hepatocytes, Kupffer cells and NK cells) we observed high correlation with the known collapsed human cell types (0.644, 0.876, 0.692 and 0.790, respectively). Correlation was low only for B cells ($r = 0.363$), consistent with the fact that expression profiles of immune cells between different species are not highly conserved. Overall, these results show that using human and mouse gene expression signatures to deconvolute cell types provides accurate cell type distributions, which correspond to the real cell type mixtures.

Additional comments, related to the revised manuscript:

1. The p-values used in Figures 2F and 3I are misleading. It looks like everything is extremely significant although the correlation coefficients are very low in most cases. This is probably due to large sample sizes. A solution might be to permute the cell compositions of each of the cell types many times, calculating the correlations of the permuted compositions with the original compositions. Then, to calculate the permutation p-value for each cell type by comparing its original correlation level to its permutations.

We agree that many p-values in the two figures are extremely significant. To improve this analysis, we permuted the samples order 1,000 times and compared the observed correlation values with the distribution of correlation values from the permutations. Many empirical p-values calculated with this method were still highly significant. For clarity, below are shown the two heatmaps (Figure 2F and Figure 3I) with asterisks indicating FDR-adjusted p-values from the permutation analysis. To address the reviewer's concern, we have removed the asterisks from the heatmaps that are present in the manuscript and added a new supplemental table (Table S4) with the permutations results, as well as additional text in the Methods (described in the section "Correlation between human and mouse cell type estimates for liver and skin").

2. Figure 4 is named figure 3 in the manuscript file.

We thank the Reviewer for pointing this out. We have corrected Figure 4 label.

3. Line 366: "By performing a permutation test, we excluded that the detection of a larger number of eQTLs using cell populations was due to the fact that a larger number of covariates was used". That's confusing - it can be interpreted as an increase in the number of eQTLs just due to the addition of power by adding covariates and not necessarily biologically meaningful covariates.

We agree with the Reviewer that this sentence was misleading. We have changed it to: "By performing a permutation test, we confirmed that the detection of a larger number of eQTLs using cell populations was biologically relevant, rather than simply resulting from the fact that a larger number of covariates were used".

Reviewers' Comments:

Reviewer #2:

Remarks to the Author:

The authors addressed all my concerns and it is now suitable for publication. It is an excellent study.